# Mitotic progression, arrest, exit or death relies on centromere structural integrity, rather than de novo transcription

Marco Novais-Cruz[1,2], Maria Alba Abad[3], Wilfred FJ van IJcken[4], Niels Galjart[5], A Arockia Jeyaprakash[3], Helder Maiato[1,2,6]*, Cristina Ferrás[1,2]*

[1]Chromosome Instability & Dynamics Laboratory, Instituto de Biologia Molecular e Celular, Universidade do Porto, Porto, Portugal; [2]Instituto de Investigação e Inovação em Saúde (i3S), Universidade do Porto, Porto, Portugal; [3]Wellcome Trust Centre for Cell Biology, University of Edinburgh, Edinburgh, United Kingdom; [4]Center for Biomics, Erasmus Medical Center, Rotterdam, Netherlands; [5]Department of Cell Biology, Erasmus Medical Center, Rotterdam, Netherlands; [6]Cell Division Group, Experimental Biology Unit, Department of Biomedicine, Faculdade de Medicina, Universidade do Porto, Porto, Portugal

**Abstract** Recent studies have challenged the prevailing dogma that transcription is repressed during mitosis. Transcription was also proposed to sustain a robust spindle assembly checkpoint (SAC) response. Here, we used live-cell imaging of human cells, RNA-seq and qPCR to investigate the requirement for de novo transcription during mitosis. Under conditions of persistently unattached kinetochores, transcription inhibition with actinomycin D, or treatment with other DNA-intercalating drugs, delocalized the chromosomal passenger complex (CPC) protein Aurora B from centromeres, compromising SAC signaling and cell fate. However, we were unable to detect significant changes in mitotic transcript levels. Moreover, inhibition of transcription independently of DNA intercalation had no effect on Aurora B centromeric localization, SAC response, mitotic progression, exit or death. Mechanistically, we show that DNA intercalating agents reduce the interaction of the CPC with nucleosomes. Thus, mitotic progression, arrest, exit or death is determined by centromere structural integrity, rather than de novo transcription.
DOI: https://doi.org/10.7554/eLife.36898.001

*For correspondence:
maiato@i3s.up.pt (HM);
cristina.ferras@ibmc.up.pt (CF)

**Competing interests:** The authors declare that no competing interests exist.

## Introduction

When cells commit to mitosis the nuclear envelope disassembles and chromatin organizes into highly condensed chromosomes. This causes the displacement of several transcription factors from DNA and the inactivation of the transcription machinery in a cell-cycle-dependent manner (*Gottesfeld and Forbes, 1997*; *Kim et al., 1997*; *Konrad, 1963*; *Martínez-Balbás et al., 1995*; *Murphy and Attardi, 1973*; *Parsons and Spencer, 1997*; *Prescott and Bender, 1962*; *Rizkallah and Hurt, 2009*; *Segil et al., 1996*; *Shermoen and O'Farrell, 1991*; *Spencer et al., 2000*; *Taylor, 1960*). Because the half-life of most transcripts exceeds the normal duration of mitosis (typically ~30 min in human cells), the prevailing dogma was that transcription is largely repressed and dispensable during mitosis in higher eukaryotes (*Gottesfeld and Forbes, 1997*; *Konrad, 1963*; *Martínez-Balbás et al., 1995*; *Murphy and Attardi, 1973*; *Prescott and Bender, 1962*; *Rizkallah and Hurt, 2009*; *Spencer et al., 2000*; *Taylor, 1960*).

Several recent studies have challenged this notion and provided evidence that the level of chromatin compaction on mitotic chromosomes is highly heterogeneous (*Nishino et al., 2012*), allowing

accessibility of some transcription factors and chromatin-modifying enzymes (*Burke et al., 2005*; *Chen et al., 2005*; *Dey et al., 2009*; *Egli et al., 2008*; *Gauthier-Rouvière et al., 1991*; *Michelotti et al., 1997*; *Segil et al., 1996*; *Yan et al., 2006*). Noteworthy, a post-translationally modified form of RNA Polymerase II that is normally associated with transcription elongation was found to bind mitotic chromosomes at their centromeres (*Chan et al., 2012*; *Dirks and Snaar, 1999*; *Liu et al., 2015*; *Molina et al., 2016*), suggesting that some transcription is still taking place during mitosis.

Transcription might also play an instrumental role during a prolonged mitosis due to incapacity to satisfy the spindle assembly checkpoint (SAC), a signaling mechanism that can delay mitosis up to several hours in the presence of unattached kinetochores (*Musacchio, 2015b*). In line with this hypothesis, the master mitotic regulator Cyclin B1 was proposed to be actively transcribed during mitosis and to be required to sustain a robust SAC response (*Mena et al., 2010*; *Sciortino et al., 2001*).

Centromere assembly, which underlies the catalytic mechanism behind the SAC, has also been recently proposed to depend on transcription of centromeric α-satellite DNA and to be regulated by non-coding RNAs (*Blower, 2016*; *Carone et al., 2009*; *Chan et al., 2012*; *Du et al., 2010*; *Grenfell et al., 2016*; *Li et al., 2008*; *Liu et al., 2015*; *Nakano et al., 2003*; *Pezer and Ugarković, 2008*; *Rošić et al., 2014*; *Topp et al., 2004*; *Wong et al., 2007*; *Zhang et al., 2005*). In particular, the centromeric localization and activation of Aurora B, the catalytic subunit of the chromosomal passenger complex (CPC) required for SAC response and error correction during mitosis (*Carmena et al., 2012*; *Santaguida et al., 2011*) was shown to be dependent on centromeric transcription (*Blower, 2016*; *Grenfell et al., 2016*). However, whether centromeric transcription persists during mitosis remains debatable.

Finally, a recent study using pulse-labeling of nascent transcripts and RNA-seq of mitotic-enriched cell populations reported that over 8000 transcripts are expressed de novo during mitosis and mitotic exit to prepare cells for the subsequent interphase (*Palozola et al., 2017*).

Although some level of transcription might still take place during mitosis, the experiments supporting this conclusion were based on the analysis of synchronized, yet impure, mitotic cell populations, fixed material and a diverse range of transcription inhibitors with imprecise temporal control of transcriptional repression. Moreover, it remains unclear whether transcription is required for normal mitotic progression and exit. For these reasons, we sought to investigate the requirement of de novo transcription during mitosis using direct live-cell imaging and a wide range of transcription inhibitors. We found that the capacity of human cells to progress, sustain, exit or die in mitosis is independent of de novo transcription. In agreement, quantitative RNA-seq and qPCR analysis failed to unveil potential transcripts produced de novo during mitosis. Finally, we uncovered that DNA-intercalating agents, which include well-established transcription inhibitors, such as actinomycin D, reduce the interaction of the CPC with nucleosomes and significantly compromise centromere structural integrity and function in a transcription-independent manner. Our findings raise awareness about the use of some transcription inhibitors in living cells and shed light on a long-lasting controversy.

## Results

### Actinomycin D compromises spindle assembly checkpoint robustness in a transcription-independent manner

We reasoned that if de novo transcription takes place during a normal mitosis, its role would be exacerbated during a prolonged mitosis. To test this, we used live-cell imaging to quantify the duration of mitosis in human HeLa cells treated with the microtubule depolymerizing drug nocodazole, which generates persistently unattached kinetochores that prevent SAC satisfaction. After 3 hr in nocodazole, we directly monitored cells that became committed to mitosis, as determined by typical cell rounding, and inhibited transcription with the well-established DNA-intercalating drug actinomycin D (*Figure 1A*). Consistent with previous observations in fixed cells (*Becker et al., 2010*), we found that live cells treated with nocodazole remained in mitosis for 19.2 ± 7.0 hr (median ± SD, n = 386 cells), whereas cells treated with nocodazole followed by actinomycin D persisted in mitosis for only 16.1 ± 5.6 hr (n = 301 cells, p<0.0001, Mann-Whitney Rank Sum Test, *Figure 1B,C* and

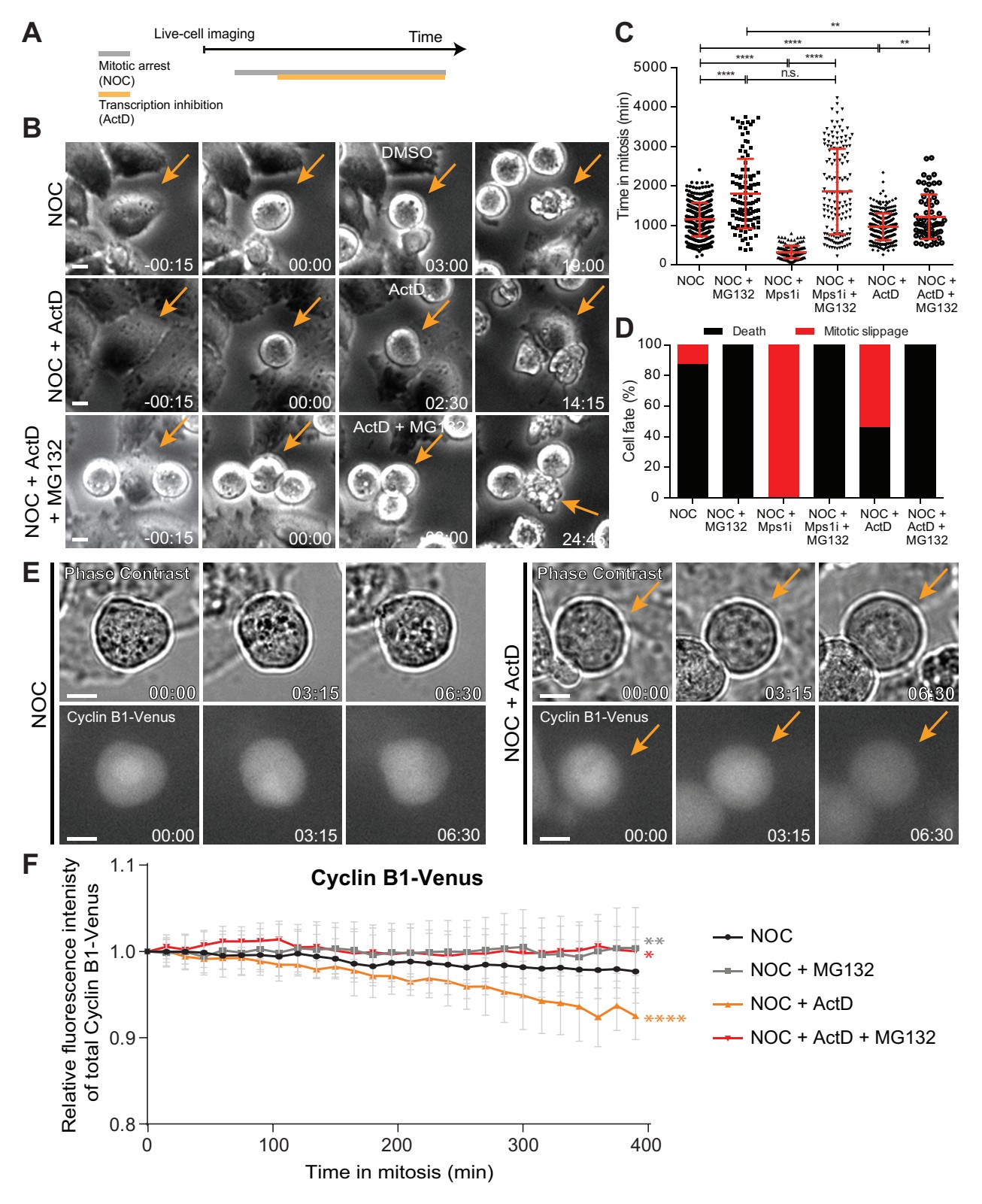

**Figure 1.** Actinomycin D compromises SAC response. (**A**) Schematic representation of the experimental protocol used to study the effect of actinomycin D (ActD) on mitotic cells by live cell imaging. (**B**) Selected time frames from phase contrast microscopy of HeLa cells treated with nocodazole and either DMSO (NOC), ActD (NOC + ActD) or ActD with MG132 (NOC + ActD + MG132). Images were acquired every 15 min. Arrows highlight examples of control (NOC) and ActD with MG132 (NOC + ActD + MG132) cells entering and dying in mitosis after a prolonged arrest. In cells

*Figure 1 continued on next page*

*Figure 1 continued*

treated with ActD (NOC + ActD), arrow highlights an example of a cell entering mitosis and then exiting mitosis through mitotic slippage. Scale bar = 10 µm. Time = hr:min. (**C**) Scattered plot showing the duration of the mitotic arrest of HeLa cells treated with nocodazole and either Mps1 inhibitor (NOC + Mps1 i) or ActD (NOC + ActD) with or without MG132. The red line represents the mean and the error bars represent the standard deviation from a pool of at least three independent experiments (NOC, 19.2 ± 7.0 hr, n = 386; NOC + MG132, 30.1 ± 14.7 hr, n = 108; NOC + Mps1 i, 5.4 ± 2.6 hr, n = 154; NOC + Mps1 i+MG132, 30.9 ± 18.1 hr, n = 150; NOC + ActD, 16.1 ± 5.6 hr, n = 301; NOC + ActD + MG132, 20.3 ± 9.4 hr, n = 65; median ±SD; \*\*p≤0.01, \*\*\*\*p≤0.0001, Mann-Whitney Rank Sum Test). (**D**) Cell fate of mitotic HeLa cells treated with the same drugs as in 1C. (**E**) Selected time frames from phase contrast and fluorescence microscopy of Cyclin B1-Venus HeLa cells treated with DMSO or ActD in the presence of nocodazole. Images were acquired every 15 min. For ActD-treated cells (NOC + ActD) arrows highlight a mitotic cell showing reduction of Cyclin B1 over time. Scale bar = 10 µm. Time = hr:min. (**F**) Cyclin B1 degradation curves for control and ActD-treated Cyclin B1-Venus HeLa cells after nocodazole treatment with or without MG132. Fluorescence intensities were normalized to the level at time = 0. The curves depict mean Cyclin B1-Venus fluorescent intensity from all analyzed cells per condition (NOC n = 12; NOC + MG132 n = 10; NOC + ActD n = 9; NOC + ActD + MG132 n = 10; from time = 0 to 26 time frames), and error bars represent the standard deviation (\*p≤0.05, \*\*p≤0.01, \*\*\*\*p≤0.0001 relative to control, Analysis of covariance).

DOI: https://doi.org/10.7554/eLife.36898.002

*Video 1*). Interestingly, approximately 90% of nocodazole-treated cells died after a prolonged mitotic delay, whereas 54% of the cells treated with nocodazole and actinomycin D underwent mitotic slippage (*Rieder and Maiato, 2004*) (*Figure 1B,D* and *Video 1*). Thus, treatment with actinomycin D compromises the capacity of cells to sustain a prolonged mitotic delay and interferes with their fate.

The relationship between the capacity to sustain a robust SAC response and the subsequent cell fate after a mitotic delay has been shown to depend on the kinetics of anaphase-promoting complex (APC)-mediated Cyclin B1 degradation, with slower degradation promoting mitotic cell death, and faster degradation promoting mitotic slippage (*Brito and Rieder, 2006*; *Gascoigne and Taylor, 2008*). We therefore tested whether inhibition of Cyclin B1 proteolysis with the proteasome inhibitor MG132 could rescue the faster exit and different fate of cells treated with nocodazole and actinomycin D. As controls, we modulated mitotic duration either by inhibition of the SAC kinase Mps1 (to accelerate mitotic exit), or inhibition of the proteasome with MG132 (to delay mitotic exit) after nocodazole treatment. Accordingly, cells treated with nocodazole and MG132 increased the mitotic delay relative to cells treated only with nocodazole (from 19.2 ± 7.0 hr to 30.1 ± 14.7 hr, p<0.0001, Mann-Whitney Rank Sum Test, *Figure 1C* and *Video 1*), and caused cell death in mitosis in 100% of the cases (*Figure 1D* and *Video 1*). Conversely, acute SAC inactivation through Mps1 inhibition after nocodazole treatment significantly decreased the mitotic delay relative to nocodazole-treated cells (from 19.2 ± 7.0 hr to 5.4 ± 2.6 hr, p<0.0001, Mann-Whitney Rank Sum Test, *Figure 1C* and *Video 1*), and caused almost immediate mitotic slippage in 100% of the cells (*Figure 1D* and *Video 1*). Proteasome inhibition with MG132 reverted the Mps1 inhibition phenotype, increasing mitotic duration and switching cell fate from mitotic slippage to death (p<0.0001, Mann-Whitney Rank Sum Test, *Figure 1C,D* and *Video 1*). Most important, the ability of actinomycin D to reduce the mitotic delay and induce slippage in cells treated with nocodazole was rescued after inhibition of proteasome activity with MG132 (p≤0.01, Mann-Whitney Rank Sum Test, *Figure 1C,D* and *Video 1*). Overall, these experiments suggest that actinomycin D compromises SAC response and the capacity to sustain a prolonged mitotic delay, in a proteasome-dependent manner.

The conclusions above are at odds with models in which SAC response depends on de novo transcription of Cyclin B1 during mitosis (*Mena et al., 2010*; *Sciortino et al., 2001*). To clarify this issue, we used quantitative time-lapse

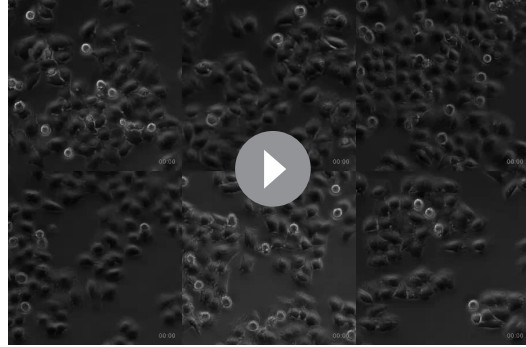

**Video 1.** Phase contrast microscopy of HeLa cells treated with DMSO, Mps1 or ActD after nocodazole (NOC) treatment with (+) or without (-) MG132. Images were acquired every 15 min. Time = h:min.

DOI: https://doi.org/10.7554/eLife.36898.003

fluorescence microscopy to monitor the levels and respective kinetics of degradation of Cyclin B1 tagged at its endogenous genomic locus with Venus (*Collin et al., 2013*) (*Figure 1E*). In agreement with previous reports, Cyclin B1-Venus fluorescence decayed very slowly in the presence of nocodazole, likely due to residual APC activity (*Figure 1F*) (*Brito and Rieder, 2006*). Treatment of nocodazole-arrested cells with actinomycin D caused a faster decline in Cyclin B1-Venus (p<0.0001, Analysis of Covariance (ANCOVA), *Figure 1F*). Interestingly, Cyclin B1-Venus levels in nocodazole-treated cells remained constant for several hours after inhibition of the proteasome with MG132, regardless of the presence of actinomycin D (p≤0.05, ANCOVA, *Figure 1F*). We concluded that actinomycin D compromises SAC robustness and the capacity to prevent APC-mediated Cyclin B1 degradation, independently of de novo transcription.

## Actinomycin D compromises the localization of Aurora B and Mad1 at centromeres/kinetochores

To determine how actinomycin D compromises SAC robustness during a prolonged mitosis, we investigated the localization of the SAC-related proteins Aurora B (at centromeres) and Mad1 (at unattached kinetochores). Accordingly, we added actinomycin D for 8 hr in cells pre-treated with nocodazole, which was kept throughout the entire duration of the experiment. To assist in the identification of mitotic cells, while assessing Aurora B activity on chromosomes, we also investigated the phosphorylation of Histone H3 at Serine 10 [pH3S10; (*Crosio et al., 2002*)]. In contrast with nocodazole-treatment in which all mitotic cells showed high levels of pH3S10, addition of actinomycin D reduced or abolished pH3S10 accumulation in approximately 17% of the nocodazole-treated cells (*Figure 2A,B*). Importantly, these cells still showed signs of chromosome condensation (as determined by DAPI staining of DNA) that were indistinguishable from neighbouring cells with normal levels of pH3S10 (*Figure 2A*). Moreover, the observed reduction/abolishment of pH3S10 was not associated with a pre-apoptotic state, as those cells were negative for a cleaved form of caspase 3 (*Figure 2A*). In line with the observed reduction/abolishment of pH3S10, these cells also showed abnormally low levels of total Aurora B and its active phosphorylated form on Threonine 232 (*Yasui et al., 2004*). Noteworthy, even in those nocodazole- and actinomycin D-treated cells that showed high levels of pH3S10, Aurora B, including its active form, was found displaced along the chromosome arms (*Figure 2A*). Similar findings were found in non-transformed RPE1 cells (*Figure 2—figure supplement 1A–B*). Thus, actinomycin D compromises normal Aurora B localization and activity.

In order to get spatiotemporal insight into the effect of actinomycin D on Aurora B localization, we monitored GFP-Aurora B by quantitative fluorescence live-cell microscopy in nocodazole-treated cells. Additionally, we further tested whether Aurora B localization was dependent on mitotic translation and on its own kinase activity by inhibiting protein synthesis with cycloheximide and treating cells with the Aurora B inhibitor ZM447439, respectively. We found that while Aurora B centromeric levels remained constant over a 5-hr period in cells treated either with nocodazole alone or nocodazole and cycloheximide, actinomycin D caused the displacement of Aurora B from centromeres onto chromosome arms 2 hr after addition of actinomycin D, leading to a 30% reduction of centromeric fluorescence after 4 hr (p<0.0001, ANCOVA, *Figure 2—figure supplement 2A,B*). Curiously, Aurora B inhibition led to an enrichment of its own levels at centromeres, without any detectable displacement onto chromosome arms (*Figure 2—figure supplement 2A,B*). These observations were confirmed by immunofluorescence analysis in fixed cells (*Figure 2C,D* and *Figure 2—figure supplement 3A–C*). These data directly demonstrate that actinomycin D displaces Aurora B from centromeres.

We then tested whether actinomycin D treatment compromised the localization of Mad1 at unattached kinetochores. We found that addition of actinomycin D for 4 hr to nocodazole-treated cells led to a 45% reduction of Mad1 at the outer kinetochore (*Figure 2C,E*). Similar findings have previously been reported for the SAC proteins Bub1 and BubR1 (*Becker et al., 2010*). Importantly, actinomycin D treatment for 4 hr did not affect the kinetochore localization of active Mps1 (Mps1 pT676; (*Jelluma et al., 2008*)) (*Figure 2—figure supplement 4A,B*) or the phosphorylation of histone H3 at threonine 3 (H3T3p), a proxy for Haspin activity previously implicated in Aurora B recruitment to centromeres (*Kelly et al., 2010*; *Wang et al., 2010*) (*Figure 2—figure supplement 4C,D*). Taken together, these experiments show that actinomycin D compromises the normal localization of

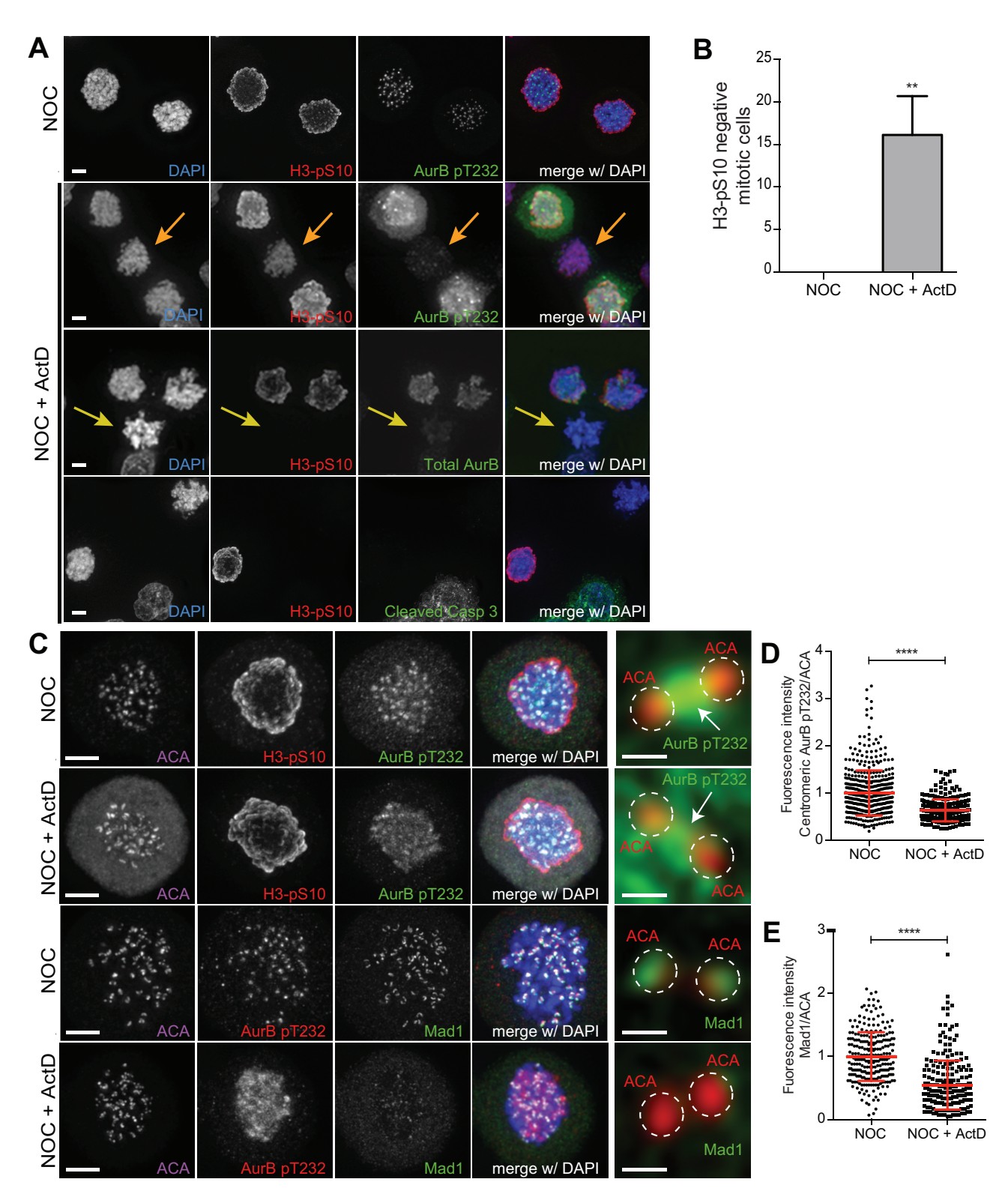

**Figure 2.** Actinomycin D compromises the localization of Aurora B and Mad1 at centromeres/kinetochores. (**A**) Representative immunofluorescence images of HeLa cells treated with DMSO (NOC) or ActD for 8 hr (NOC + ActD) after nocodazole treatment with the indicated antibodies. In the NOC + ActD condition orange arrows highlight a mitotic cell with reduced levels of phosphorylated Histone H3 (H3–pS10) and devoid of the active form of Aurora B (AurB pT232). Yellow arrows highlight a mitotic cell with abolished levels of phospho H3 (H3–pS10) and devoid of total Aurora B. Scale
*Figure 2 continued on next page*

*Figure 2 continued*

bar = 5 µm. (**B**) Percentage of mitotic phospho H3-negative cells in control (NOC) and 8 hr actinomycin D (NOC + ActD)-treated HeLa cells. The bar graph represents the mean and the standard deviation from three independent experiments (NOC 0.0 ± 0.0, n = 600; NOC + ActD, 16.1 ± 4.5, n = 600; **p≤0.01 relative to control, t test) (**C**) Representative immunofluorescence images of Hela cells treated with DMSO (NOC) or ActD for 4 hr (NOC + ActD) after nocodazole treatment with the indicated antibodies. Scale bar = 5 µm. 10x magnification of a pair of kinetochores are shown in the right. Dashed circle encompasses a single kinetochore and the arrow indicates the position of centromeric Aurora B. Scale bar = 0.5 µm. (**D**) Normalized ratio of pAurora B/ACA fluorescence signal at inner centromere of NOC and NOC + ActD. Each dot represents an individual kinetochore. The red line represents the mean of all quantified kinetochores and the error bars represent the standard deviation from a pool of at least two independent experiments. (NOC, 1.00 ± 0.48, n = 474; NOC + ActD, 0.63 ± 0.23, n = 309; ****p≤0.0001 relative to control, Mann-Whitney Rank Sum Test). (**E**) Normalized ratio of Mad1/ACA fluorescence signal at outer kinetochores of NOC and NOC + ActD cells. Each dot represents an individual kinetochore. The red line represents the mean of all quantified kinetochores and the error bars represent the standard deviation from a pool of at least two independent experiments. (NOC, 1.00 ± 0.39, n = 261; NOC + ActD, 0.55 ± 0.39, n = 258; ****p≤0.0001 relative to control, Mann-Whitney Rank Sum Test).

DOI: https://doi.org/10.7554/eLife.36898.004

The following figure supplements are available for figure 2:

**Figure supplement 1.** Actinomycin D compromise Aurora B localization in RPE1 cells.

DOI: https://doi.org/10.7554/eLife.36898.005

**Figure supplement 2.** Actinomycin D causes Aurora B displacement from centromeres onto chromosome arms.

DOI: https://doi.org/10.7554/eLife.36898.006

**Figure supplement 3.** Inhibition of mitotic translation or Aurora B kinase activity during prometaphase does not affect Aurora B localization.

DOI: https://doi.org/10.7554/eLife.36898.007

**Figure supplement 4.** Actinomycin D does not compromise pMPS1 localization at kinetochores and pH3T3 levels.

DOI: https://doi.org/10.7554/eLife.36898.008

Aurora B at centromeres and SAC proteins to unattached kinetochores, independently of an effect over Mps1 and Haspin kinase activity.

## Actinomycin D or Aurora B inhibition disturbs the kinetochore recruitment of Knl1 and Mad1

Next, we investigated whether the reduction of Mad1 at unattached kinetochores in the presence of actinomycin D was induced by the perturbation of Aurora B localization and activity at centromeres. To do so, we started by evaluating the status of known Aurora B substrates within the KMN network (*Welburn et al., 2010*) after actinomycin D treatment. Consistent with an attenuation of Aurora B activity at centromeres, we found that addition of actinomycin D for 4 hr to nocodazole-treated cells led to a 22% and 43% reduction on the phosphorylation levels of Knl1 and Dsn1 (a member of the Mis12 complex), respectively (*Figure 3A–D*). Interestingly, we also found a 29% reduction in total Knl1 (*Figure 3E,F*), whereas total Dsn1, Hec1 (a member of the Ndc80 complex) and CENP-A were unaffected by actinomycin D treatment (*Figure 3G,H*, *Figure 3—figure supplement 1A–D*). Thus, perturbation of Aurora B centromeric localization by actinomycin D translates into a local reduction of activity and the consequent decrease in phosphorylation of key Aurora B substrates at the kinetochore. Moreover, while actinomycin D treatment for 4 hr does not perturb the normal localization of CENP-A and members of the Ndc80 and Mis12 complex, it compromises the recruitment of Knl1 to unattached kinetochores.

Because Knl1 is known to recruit several SAC proteins (including Mad1) to kinetochores (*Musacchio, 2015a*), we tested whether Knl1 and Mad1 recruitment to kinetochores depends on Aurora B activity at centromeres. We found that, similar to actinomycin D treatment, Aurora B inhibition in nocodazole-treated cells caused a 38% and 28% reduction of Mad1 and Knl1 at unattached kinetochores, respectively (*Figure 3—figure supplement 2A–D*). Importantly, it has previously been shown that constitutive targeting of Aurora B to centromeres by expressing a Cenp-B-INCENP fusion protein was able to rescue SAC response in actinomycin D-treated cells (*Becker et al., 2010*). Overall, these data suggest that perturbation of Aurora B localization and consequent decrease in activity at centromeres is the primary effect of Actinomycin D treatment and this likely accounts for the weakened SAC response due to a downstream effect over Knl1 and Mad1 recruitment.

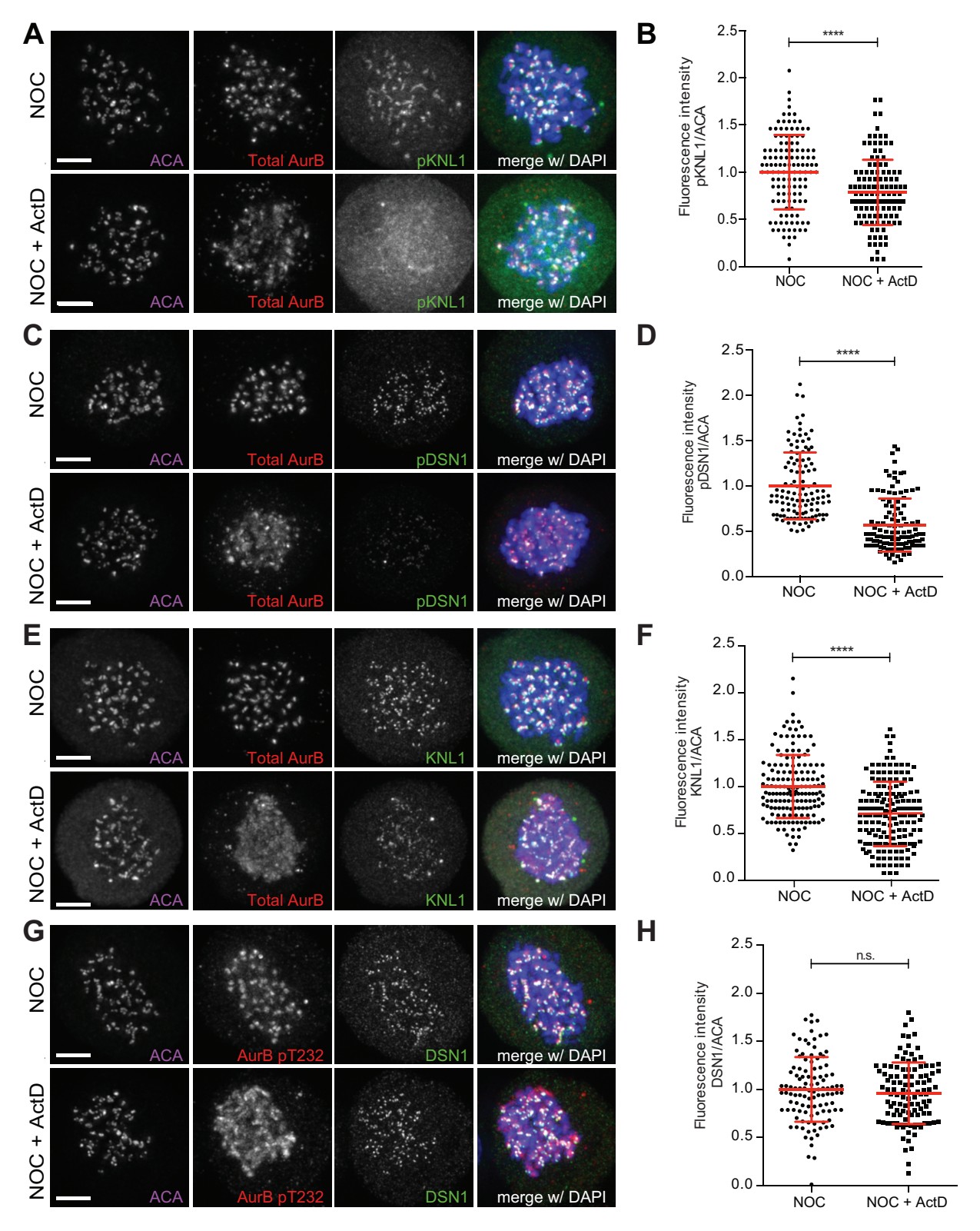

**Figure 3.** Actinomycin D affect recruitment of KNL1 and pDSN1 to kinetochores. (**A**) Representative immunofluorescence images of HeLa cells treated with DMSO (NOC) or ActD for 4 hr (NOC + ActD) after nocodazole treatment with the indicated antibodies. Scale bar = 5 μm. (**B**) Normalized ratio of pKNL1/ACA fluorescence signal at outer kinetochores of NOC and NOC + ActD cells. Each dot represents an individual kinetochore. The red line represents the mean of all quantified kinetochores and the error bars represent the standard deviation from a pool of two technical replicates (NOC,

*Figure 3 continued on next page*

Figure 3 continued

1.00 ± 0.40, n = 126; NOC + ActD, 0.79 ± 0.35, n = 126; ****p≤0.0001 relative to control, t test). (C) Representative immunofluorescence images of Hela cells treated with DMSO (NOC) or ActD for 4 hr (NOC + ActD) after nocodazole treatment with the indicated antibodies. Scale bar = 5 µm. (D) Normalized ratio of pDNS1/ACA fluorescence signal at outer kinetochores of NOC and NOC + ActD cells. Each dot represents an individual kinetochore. The red line represents the mean of all quantified kinetochores and the error bars represent the standard deviation from a pool of two technical replicates (NOC, 1.00 ± 0.37, n = 116; NOC + ActD, 0.57 ± 0.29, n = 116; ****p≤0.0001 relative to control, Mann-Whitney Rank Sum Test). (E) Representative immunofluorescence images of HeLa cells treated with DMSO (NOC) or ActD for 4 hr (NOC + ActD) after nocodazole treatment with the indicated antibodies. Scale bar = 5 µm. (F) Normalized ratio of KNL1/ACA fluorescence signal at outer kinetochores of NOC and NOC + ActD cells. Each dot represents an individual kinetochore. The red line represents the mean of all quantified kinetochores and the error bars represent the standard deviation from a pool of two technical replicates (NOC, 1.00 ± 0.34, n = 160; NOC + ActD, 0.71 ± 0.34, n = 160; ****p≤0.0001 relative to control, Mann-Whitney Rank Sum Test). (G) Representative immunofluorescence images of HeLa cells treated with DMSO (NOC) or ActD for 4 hr (NOC + ActD) after nocodazole treatment with the indicated antibodies. Scale bar = 5 µm. (H) Normalized ratio of DSN1/ACA fluorescence signal at outer kinetochores of NOC and NOC + ActD cells. Each dot represents an individual kinetochore. The red line represents the mean of all quantified kinetochores and the error bars represent the standard deviation from a pool of two technical replicates (NOC, 1.00 ± 0.34, n = 110; NOC + ActD, 0.96 ± 0.32, n = 110; n.s. p>0.05 relative to control, t test).

DOI: https://doi.org/10.7554/eLife.36898.009

The following figure supplements are available for figure 3:

**Figure supplement 1.** Actinomycin D does not affect levels of HEC1 and CENP-A (A) Representative immunofluorescence images of HeLa cells treated with DMSO (NOC) or ActD for 4 hr (NOC + ActD) after nocodazole treatment with the indicated antibodies.
DOI: https://doi.org/10.7554/eLife.36898.010

**Figure supplement 2.** Aurora B inhibition affects recruitment of Mad1 and KNL1 to kinetochores.
DOI: https://doi.org/10.7554/eLife.36898.011

## Actinomycin D does not affect total CPC protein levels nor the levels of mitotic transcripts

Western blot analysis revealed that Aurora B, as well as the total protein levels of the CPC regulatory subunits Survivin and Borealin remained unchanged relative to controls (including inhibition of translation during mitosis with cycloheximide or Aurora B inhibition with ZM447439) after 4 hr treatment with actinomycin D (*Figure 4A–C*). To test whether Aurora B was regulated by an effect of actinomycin D over mitotic transcription, we performed a comparative genome-wide RNA-seq analysis between cells treated either with nocodazole alone or nocodazole and actinomycin D. Importantly, only mitotic cells were analyzed, as they were obtained by shake-off after nocodazole treatment and subsequently transferred to new culture flasks before addition of actinomycin D (*Figure 4A*). Our quantitative transcriptome analysis did not reveal any significant changes either in annotated or novel assembled transcripts (including all biotypes) upon actinomycin D treatment, ruling out differential gene expression after transcription inhibition during mitosis. Importantly, we confirmed that Cyclin B1, as well as all key mitotic regulators, including Aurora B, were not de novo transcribed even during a prolonged mitosis (*Figure 4D*; see Materials and methods).

Due to the low mappability of the centromere, RNA-seq reads could not be properly aligned to these regions. Consequently, the role of centromeric transcription, namely of regulatory non-coding RNAs, could not be properly assessed in the previous experiment. Nevertheless, we were able to directly measure by qPCR the expression of centromeric α-satellite RNA relative to a control housekeeping gene (GAPDH) in cells that had been arrested in mitosis with nocodazole, in the presence or absence of actinomycin D. We found that actinomycin D treatment did not reduce the level of centromeric α-satellite RNA (*Figure 4E,F*). Taken together, our results demonstrate that inhibition of transcription with actinomycin D during a prolonged mitosis caused by persistently unattached kinetochores does not impair normal transcript levels, including centromeric α-satellite RNA, nor the normal levels of CPC proteins.

## Inhibition of transcription independently of DNA-intercalation does not affect spindle assembly checkpoint robustness nor the capacity to trigger mitotic cell death

So far, our results indicated that actinomycin D compromises SAC response and Aurora B centromeric localization independently of de novo transcription. Therefore, we reasoned that actinomycin D could displace Aurora B from centromeres, not by inhibiting mitotic transcription, but through its

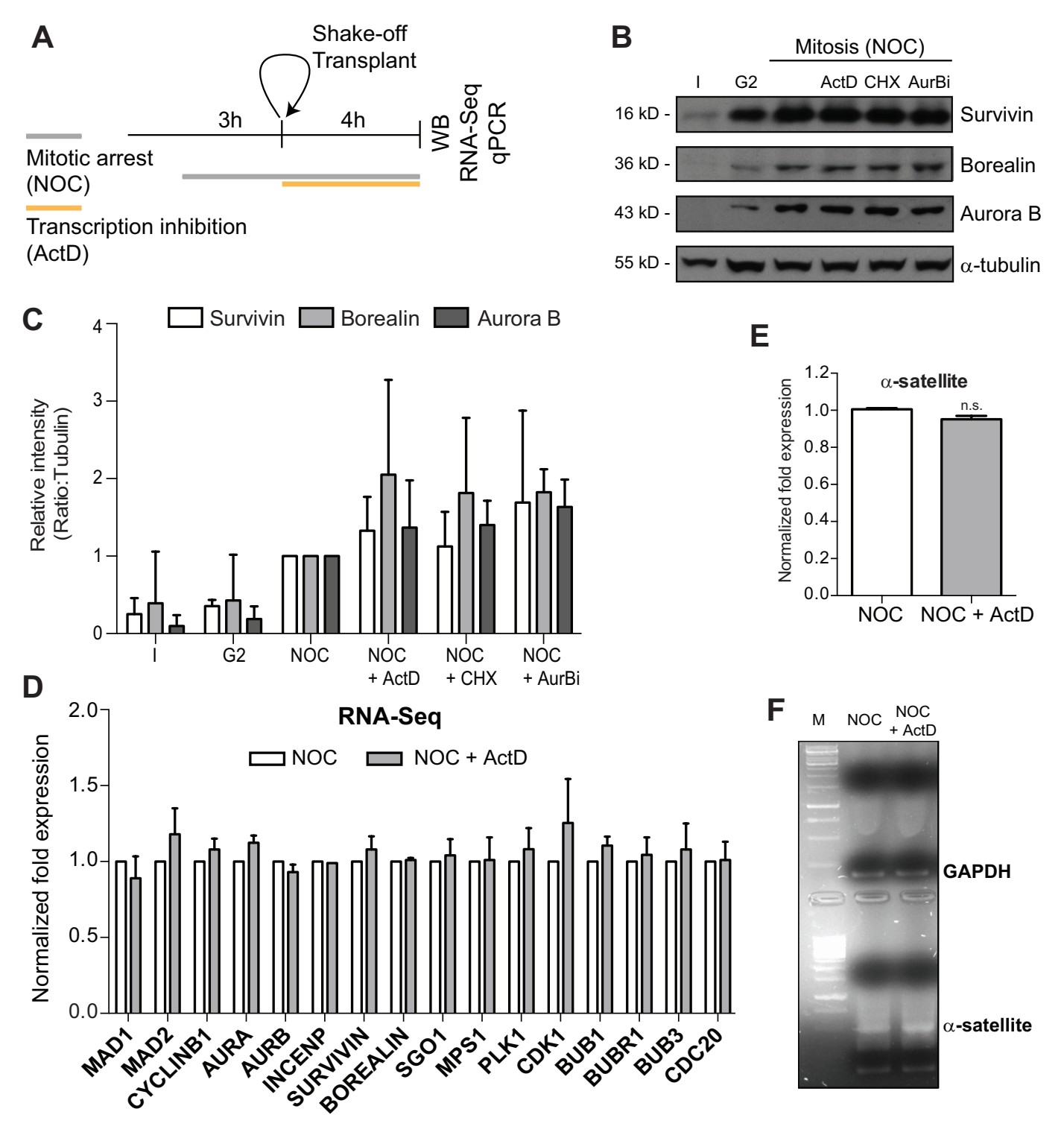

**Figure 4.** Actinomycin D treatment does not affect the levels of Aurora B or mitotic transcripts. (**A**) Schematic representation of the experimental protocol used to study the effect of ActD in NOC induced mitotic arrested cells by western blot, RNA-seq and qPCR. (**B**) Western blot analysis of asynchronous (I), G2, and mitotic HeLa cell extracts (Mitosis) with the indicated antibodies. Mitotic extracts were derived from isolated mitotic cells treated cells with DMSO (control), ActD (ActD), cycloheximide (CHX) or Aurora B inhibitor (AurBi). α-tubulin was used as loading control. Approximate molecular weights are shown on the left. (**C**) Quantification of western blot is depicted in B. The bar represents the mean and the standard deviation from three independent experiments. (**D**) Normalized expression after RNA-seq of the indicated genes in control or ActD-treated mitotic cells. Error

*Figure 4 continued on next page*

*Figure 4 continued*

bars represent standard deviations from two independent experiments. (E) Normalized expression after qPCR of human alpha satellite gene in control or ActD-treated mitotic cells (NOC, 1.00 ± 0.01; NOC + ActD, 0.95 ± 0.02, median ±SD from three technical replicates, n.s. p>0.05 relative to control, Mann-Whitney Rank Sum Test). (F) Agarose gel electrophoresis of the qPCR product after amplification of cDNA, obtained from control or ActD-treated mitotic cells, with primers specific for the human alpha-satellite and GAPDH.

DOI: https://doi.org/10.7554/eLife.36898.012

ability to intercalate with DNA. In line with this hypothesis, it was previously shown that actinomycin D, but not α-amanitin [a potent and selective inhibitor of RNA polymerase II that triggers the degradation of the largest RNA polymerase II subunit (*Nguyen et al., 1996*), interferes with the normal centromeric localization of Aurora B (*Becker et al., 2010*). However, α-amanitin is a slow-uptake drug and inhibition of transcription was not confirmed in these studies. For this reason, we first compared the effect of α-amanitin with that of actinomycin D or triptolide, a fast and irreversible transcription inhibitor that prevents the formation of the 'transcription bubble' without intercalating with DNA (*Titov et al., 2011*; *Vispé et al., 2009*). To determine the extent of transcription inhibition by these drugs, we used qPCR to amplify Mcl1, a short half-life transcript isolated from asynchronous cells after 4 hr treatment either with α-amanitin, actinomycin D or triptolide. As suspected, we could not detect any significant effect of α-amanitin over Mcl1 transcript levels relative to controls (*Figure 5A*). In sharp contrast, treatment with either actinomycin D or triptolide resulted in a 5-fold decrease in Mcl1 transcript levels (*Figure 5A*) and led interphase cells to death within 4–5 hr in the presence of either actinomycin D or triptolide (*Videos 1* and *2*), suggesting an efficient and equivalent inhibition of transcription in both scenarios. Finally, we confirmed that, within the time course of our experiment, triptolide, but not α-amanitin, caused the degradation of the large subunit of RNA Polymerase II (*Figure 5—figure supplement 1A*). Thus, triptolide, but not α-amanitin, efficiently inhibits transcription in living cells and it does so independently of DNA intercalation.

Next, we investigated whether transcription inhibition with triptolide compromises SAC robustness in cells treated with nocodazole. We found no significant differences in mitotic duration between nocodazole-treated cells with or without triptolide (19.3 ± 7.1 hr vs 19.5 ± 5.8 hr, respectively, median ±SD, n = 301 cells, p=0.65, Mann-Whitney Rank Sum Test, *Figure 5B* and *Video 2*). Moreover, contrary to what was previously observed in the case of actinomycin D, we did not detect any effect of triptolide over cell fate after the mitotic delay imposed by nocodazole treatment and most cells died in mitosis (*Figure 5C* and *Video 2*). Triptolide also did not cause the displacement of Aurora B from centromeres in both fixed and live-cell assays (*Figure 5D–G*). These results, supported by our RNA-seq data, indicate that SAC robustness, Aurora B centromeric localization, as well as the capacity to die in mitosis after a prolonged mitotic delay is independent of de novo transcription.

## DNA intercalation interferes with the interaction between the CPC and nucleosomes

Our previous results suggest that DNA intercalation, rather than transcriptional inhibition, causes the displacement of Aurora B from centromeres. To test this, we took advantage from another bona fide DNA intercalating compound, ethidium bromide, which has not been implicated in transcription inhibition (*Bensaude, 2011*). Indeed, we found that interphase cells were able to enter mitosis after treatment with ethidium bromide (*Video 2*). Additionally, Mcl1 transcript levels were only slightly decreased (without statistic significance) after 4 hr of ethidium bromide treatment (*Figure 6A*). In agreement, cells treated with nocodazole and ethidium bromide spent slightly less time in mitosis when compared to cells treated with nocodazole only, but the difference was not statistically relevant (19.2 ± 7.0 hr vs 18.1 ± 5.9 hr, median ±SD, n = 386;151 cells/condition, p=0.13, Mann-Whitney Rank Sum Test, *Figure 6B* and *Video 2*). However, similar to actinomycin D, ethidium bromide caused a significant effect over cell fate after the mitotic delay imposed by nocodazole, causing mitotic slippage in approximately 50% of the cells (*Figure 6C* and *Video 2*). Ethidium bromide also displaced Aurora B from centromeres in fixed cells, although to a less extent when compared with actinomycin D (*Figure 6D,E*). These data suggest that DNA intercalating drugs interfere with normal Aurora B localization at centromeres and compromise cell fate after a prolonged mitosis.

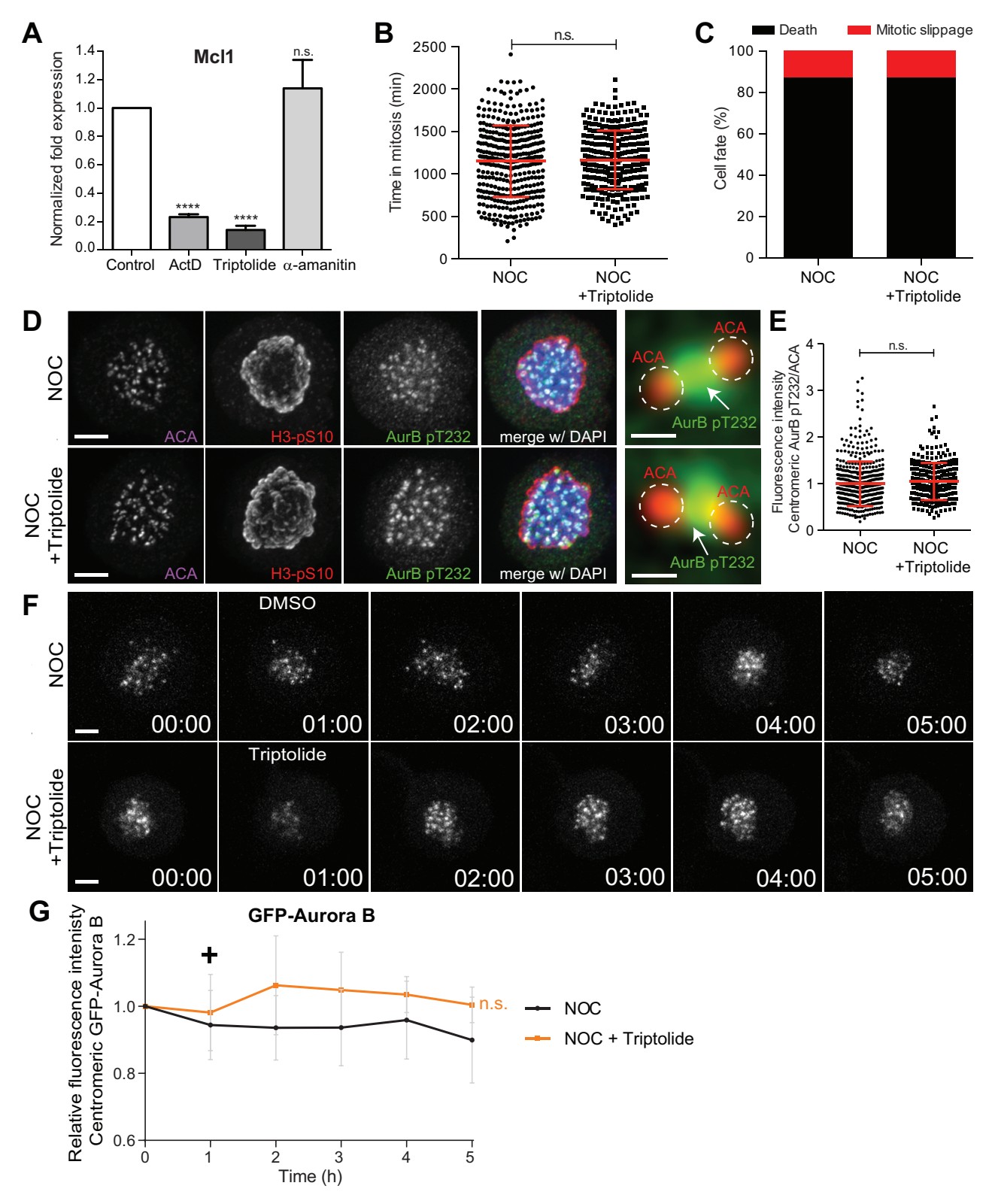

**Figure 5.** Inhibition of transcription independently of DNA-intercalation does not affect spindle assembly checkpoint robustness nor the capacity to trigger mitotic cell death. (**A**) Normalized expression of Mcl1 in asynchronous control HeLa cells or cells treated with ActD, triptolide or α-amanitin (ActD, 0.23 ± 0.02, Triptolide, 0.14 ± 0.03; α-amanitin, 1.13 ± 0.2, median ±SD from three technical replicates, n.s. p>0.05, ****p≤0.0001 relative to control, t test). (**B**) Scatter plot showing the duration of the mitotic arrest of control (NOC) and HeLa cells treated with Triptolide (NOC + Triptolide)

*Figure 5 continued on next page*

*Figure 5 continued*

after nocodazole treatment. The red line represents the mean and the error bars represent the standard deviation from a pool of at least two independent experiments. (NOC, 19.2 ± 7.0 hr, n = 386; NOC + Triptolide, 30.1 ± 14.7 hr, n = 301; median ±SD; n.s. p>0.05, Mann-Whitney Rank Sum Test) (C) Cell fate of mitotic HeLa cells treated with the same drugs as in 4B. (D) Representative immunofluorescence images of HeLa cells treated with DMSO (NOC) or Triptolide for 4 hr (NOC + Triptolide) after nocodazole treatment with the indicated antibodies. Scale bar = 5 µm. 10x magnification of a pair of kinetochores are shown in the right. Dashed circle encompasses a single kinetochore and the arrow indicates the position of centromeric Aurora B. Scale bar = 0.5 µm. (E) Normalized ratio of pAurora B/ACA fluorescence signal at inner centromere of NOC and NOC + Triptolide. Each dot represents an individual kinetochore. The red line represents the mean of all quantified kinetochores and the error bars represent the standard deviation from a pool of two independent experiments. (NOC, 1.00 ± 0.48, n = 474; NOC + Triptolide, 1.05 ± 0.00.40, n = 296; n.s. p>0.05 relative to control, Mann-Whitney Rank Sum Test) (F) Selected time frames from fluorescence microscopy of HeLa LAP-Aurora B cells treated with DMSO or Triptolide after nocodazole treatment. Images were acquired every 10 min. Scale bar = 5 µm. Time = h:min. (G) Relative fluorescence intensity curves of centromeric GFP-Aurora B for control and Triptolide-treated HeLa cells after nocodazole treatment. Fluorescence intensities were normalized to the level at time = 0. The curves depict mean GFP-Aurora B fluorescent intensity from analyzed cells (NOC n = 9; NOC + Triptolide n = 8; from time = 0 to 30 time frames), and error bars represent the standard deviation (n.s. p>0.05 relative to control, Analysis of covariance). Plus symbol on the graph indicates the time when the drug was added.

DOI: https://doi.org/10.7554/eLife.36898.013

The following figure supplement is available for figure 5:

**Figure supplement 1.** RNA Pol II is not degraded with 12 hr α-amanitin treatment.
DOI: https://doi.org/10.7554/eLife.36898.014

To directly test whether DNA intercalating drugs interfere with the capacity of Aurora B to associate with centromeric chromatin, we probed the ability of the CPC Centromere Targeting Module (CPC-CTM), composed by full-length Borealin, full-length Survivin and INCENP 1–58, to bind recombinant nucleosome core particles (NCPs) by means of Electrophoretic Mobility Shift Assays (EMSA), in the presence or absence of actinomycin D, ethidium bromide or triptolide (*Figure 6F*). Consistent with our fixed and live-cell data, actinomycin D or ethidium bromide-treated NCPs showed ~50% reduction in CPC-CTM binding, when compared to DMSO- or Triptolide-treated NCPs (p≤0.05, t test, *Figure 6G*). Thus, DNA intercalating drugs interfere with centromere integrity.

## The capacity to satisfy the spindle assembly checkpoint and resume mitosis and cytokinesis after a prolonged mitotic delay does not require de novo transcription

Lastly, we tested whether the capacity to progress and exit mitosis requires de novo mitotic transcription. To assure an efficient transcriptional repression during mitosis, we performed a nocodazole washout experiment. Accordingly, HeLa cells were incubated for 5 hr with nocodazole and the transcription inhibitors (actinomycin D or triptolide) were added in the last 2 hr of the mitotic blockage. Nocodazole was subsequently washed out and the cells were allowed to progress through mitosis only in the presence of the transcription inhibitors (*Figure 7A,B* and *Video 3*). Live-cell imaging was used to directly determine the duration of mitosis, while assessing cell fate. We found that control and triptolide-treated cells took a similar time to exit mitosis after nocodazole washout (79.1 ± 28.7 min vs 86.2 ± 31.7 min, median ± SD, n = 116;113 cells/condition, p=0.11, Mann-Whitney Rank Sum Test, *Figure 7C* and *Video 3*) without any apparent defect (*Figure 7D* and *Video 3*).

In contrast, actinomycin D-treated cells showed a significant delay in mitotic exit (*Figure 7C* and *Video 3*) and underwent mitotic slippage (33%) or failed cytokinesis (47%) (*Figure 7D* and *Video 3*). Close inspection of actinomycin D-treated cells after nocodazole washout by immunofluorescence revealed persistent DNA bridges at the cleavage furrow (*Figure 7—figure supplement 1A*), which might have accounted for the observed failure in completing cytokinesis. Taken together, these experiments indicate that the capacity of human cells to progress and

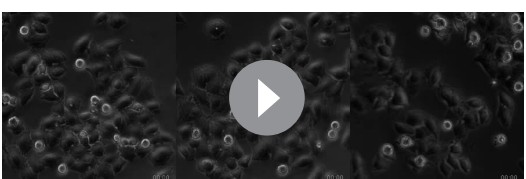

**Video 2.** Phase contrast microscopy of HeLa cells treated with DMSO, Triptolide or ethidium bromide after nocodazole (NOC) treatment. Images were acquired every 15 min. Time = h:min.
DOI: https://doi.org/10.7554/eLife.36898.015

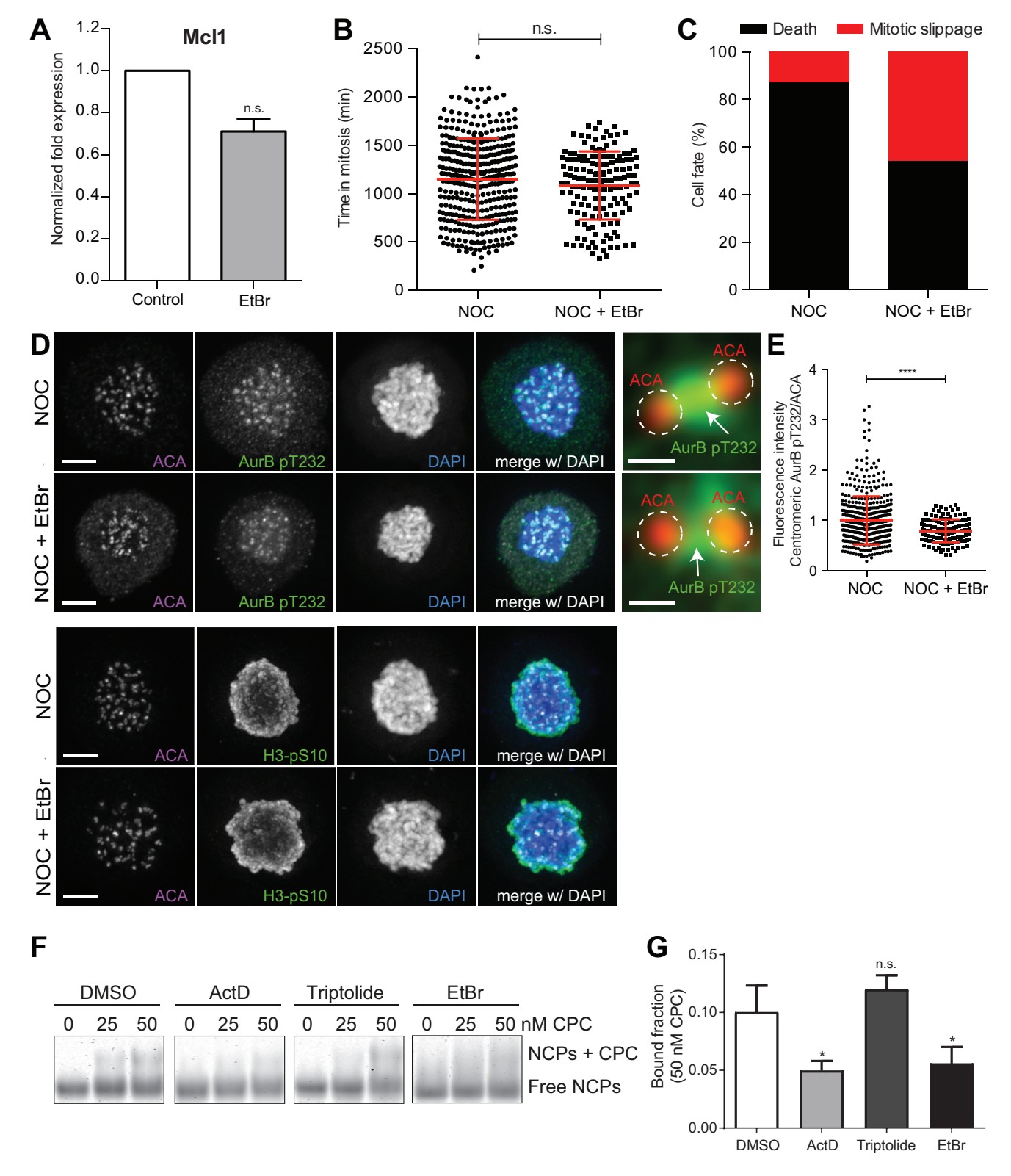

**Figure 6.** DNA intercalation compromises SAC response by interfering with Aurora B localization. (**A**) Normalized expression of Mcl1 in asynchronous control HeLa cells or cells treated with ethidium bromide (EtBr) (EtBr, 0.71 ± 0.06, median ±SD from three technical replicates, n.s. p>0.05, Mann-Whitney Rank Sum Test). (**B**) Scattered plot showing the duration of the mitotic arrest of control (NOC) and Hela cells treated with ethidium bromide (NOC + EtBr) after nocodazole treatment (NOC, 19.2 ± 7.0 hr, n = 386; NOC + EtBr, 18.1 ± 5.9 hr, n = 151; median ±SD; n.s. p>0.05, Mann-Whitney

*Figure 6 continued on next page*

*Figure 6 continued*

Rank Sum Test). The red line represents the mean and the error bars represent the standard deviation from a pool of at least two independent experiments. (C) Cell fate of mitotic HeLa cells treated with the same drugs as in 5B. (D) Representative immunofluorescence images of HeLa cells treated with DMSO (NOC) or ethidium bromide for 4 hr (NOC + EtBr) after nocodazole treatment with the indicated antibodies. Scale bar = 5 μm. 10x magnification of a pair of kinetochores are shown in the right. Dashed circle encompasses a single kinetochore and the arrow indicate the position of centromeric Aurora B. Scale bar = 0.5 μm. (E) Normalized ratio of pAurora B/ACA fluorescence signal at inner centromere of NOC and NOC + EtBr. Each dot represents an individual kinetochore. The red line represents the mean of all quantified kinetochores and the error bars represent the standard deviation. (NOC, 1.00 ± 0.48, n = 474; NOC + EtBr, 0.79 ± 0.23, n = 143; ****p≤0.0001 relative to control, Mann-Whitney Rank Sum Test). (F) Shows the Electrophoretic Mobility Shift Assay (EMSA) experiments carried out to assess the binding chromosomal passenger complex (CPC) ability to recombinant nucleosome core particles (NCPs) in the presence of DMSO, ActD, Triptolide or EtBr. Different molar concentrations (0, 25 and 50 nM) of CPC were tested. (G) The bar diagram shows the densitometric profile of the autoradiograph depicted in F. The bar represents the mean ±SD from at least three independent experiments (n.s. p>0.05, *p≤0.05 relative to control, t test).

DOI: https://doi.org/10.7554/eLife.36898.016

exit mitosis, including the completion of cytokinesis, is independent of de novo transcription.

## Discussion

Whether transcription is permissive and required during mitosis is a recurrent debate in the field. The resolution of this long-lasting controversy has been hampered by the difficulty to efficiently and specifically repress transcription during mitosis, which often takes only ~30 min in human cells. It has, however, been proposed that the master mitotic regulator Cyclin B1 is actively transcribed during mitosis (*Sciortino et al., 2001*) and de novo transcription is required to sustain a mitotic delay under conditions that prevent SAC satisfaction for several hours (*Mena et al., 2010*). Centromere/kinetochore assembly has also been linked to active transcription (*Bobkov et al., 2018*; *Carone et al., 2009*; *Chan et al., 2012*; *Du et al., 2010*; *Grenfell et al., 2016*; *Li et al., 2008*; *Liu et al., 2015*; *Nakano et al., 2003*; *Pezer and Ugarković, 2008*; *Rošić et al., 2014*; *Topp et al., 2004*; *Wong et al., 2007*; *Zhang et al., 2005*), including the localization and activity of Aurora B, as shown in *Xenopus* egg extracts (*Blower, 2016*; *Grenfell et al., 2016*). More recently, an entire transcription program was proposed to remain constitutively active during mitosis and mitotic exit in human cells (*Palozola et al., 2017*; *Strzyz, 2017*; *Timmers and Verrijzer, 2017*), but whether this potential program is required for mitotic progression and exit was not elucidated.

By combining direct live-cell imaging, while monitoring the efficiency and specificity of transcription inhibition at the whole genome level, we show that the capacity of human cells to sustain, progress, exit or die in mitosis does not require de novo transcription. Moreover, we demonstrate that commonly used transcription inhibitors, such as actinomycin D and α-amanitin show serious limitations in live-cell experiments that aim to understand mitosis. Actinomycin D and other DNA intercalating agents caused partial dissociation of the CPC from nucleosomes, thereby compromising Aurora B centromeric localization and SAC response. On the other hand, the slow uptake drug α-amanitin failed to efficiently inhibit transcription even after several hours in mitosis. Most importantly, efficient inhibition of mitotic transcription independently of DNA intercalation using triptolide had no discernible effect over Aurora B centromeric localization or SAC response. We concluded that centromere integrity, rather than mitotic transcription, is required for proper localization and activity of Aurora B and to mount a robust SAC able to sustain mitosis in human cells for several hours in the event of unattached kinetochores. These findings are consistent with a role of Aurora B in the SAC under conditions that prevent microtubule attachment (*Santaguida et al., 2011*) and our work offers a possible explanation for such a role. Accordingly, we showed that both Aurora B activity and its stable association with centromeres are important for normal Knl1 and Mad1 recruitment to unattached kinetochores. However, at this stage, we cannot formally exclude other effects caused by a prolonged actinomycin D treatment under conditions that prevent SAC satisfaction.

In light of a recent study reporting the identification of over 900 nascent transcripts from allegedly metaphase cells (*Palozola et al., 2017*), it was surprising that our genome wide RNA-seq or qPCR analyses failed to reveal any significant alteration in gene expression, including Cyclin B1 and other mitotic genes, during a prolonged mitosis after transcription inhibition with actinomycin D. However, in the previous study, nocodazole-treated cells were only 95% pure and transcripts

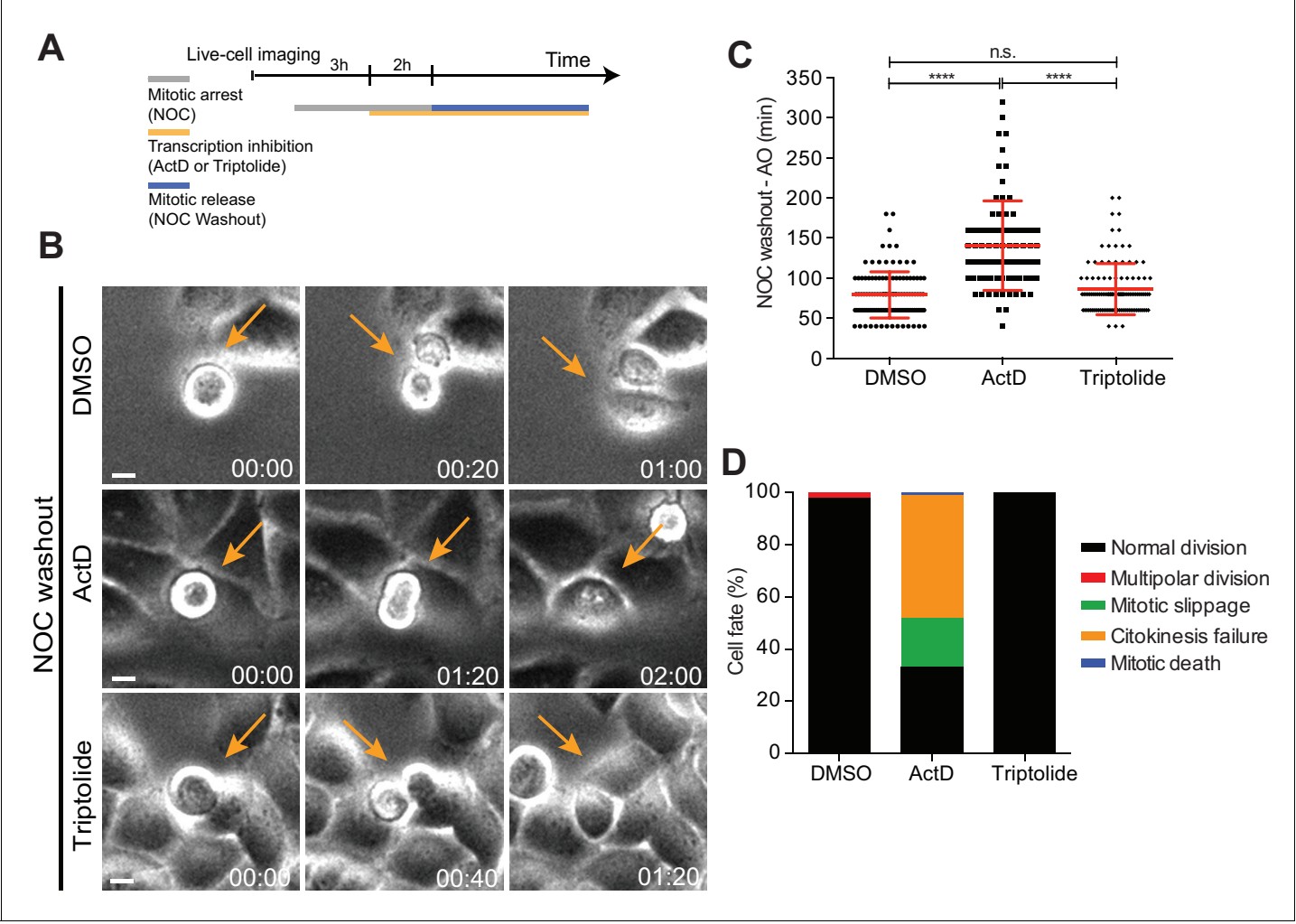

**Figure 7.** Mitotic progression and cytokinesis after nocodazole washout does not require de novo transcription. (**A**) Schematic representation of the experimental protocol used to study the effect of the transcription inhibitors ActD and triptolide on mitotic progression. (**B**) Selected time frames from phase contrast microscopy of HeLa cells treated after the NOC washout with DMSO, ActD or Triptolide. Images were acquired every 20 min. For control (DMSO) and Triptolide (Triptolide), arrows highlight a normal mitotic progression after nocodazole washout. For ActD-treated cells (ActD), arrows highlight a cell that fails to divide after release from nocodazole arrest. Scale bar = 10 μm. Time = hr:min. (**C**) Scattered plot showing the mitotic duration between nocodazole washout and anaphase onset in control (DMSO), ActD and Triptolide treated HeLa cells (DMSO, 79.1 ± 28.8 min, n = 116; ActD, 140.5 ± 55.7 hr, n = 79; Triptolide, 86.2 ± 31.9 hr, n = 113, median ±SD from a pool of two technical replicates, n.s. p>0.05, ****p≤0.0001, Mann-Whitney Rank Sum Test). (**D**) Cell fate of mitotic HeLa cells treated with the same drugs as in 7B.

DOI: https://doi.org/10.7554/eLife.36898.017

The following figure supplement is available for figure 7:

**Figure supplement 1.** Actinomycin D promotes cytokinesis failure.

DOI: https://doi.org/10.7554/eLife.36898.018

isolated 40 min after nocodazole washout might have derived from cells that had already exited mitosis and reached early G1 stage where transcription is expected to be permissive (*Hsiung et al., 2016*). Thus, the use of synchronized, yet impure, cell populations, as opposed to monitoring transcription inhibition only after cells commit to mitosis by direct live-cell imaging, might account for the differences observed between studies. Nevertheless, it remains possible that, as cells progress through mitosis and enter anaphase, de novo transcription starts to be permissive, despite its dispensability for the completion of and exit from mitosis.

Consistent with a global shut-down of transcription during prometaphase (the physiological equivalent of nocodazole-arrested cells), RNA Polymerase II is generally found displaced from

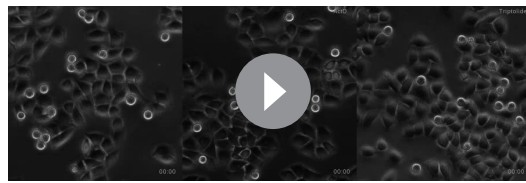

**Video 3.** Phase contrast microscopy of HeLa cells treated with DMSO, ActD or Triptolide after NOC washout. Images were acquired every 20 min. Time = hr:min.
DOI: https://doi.org/10.7554/eLife.36898.019

chromatin (*Hsiung et al., 2016*), with the notorious exception of the centromeric region (*Chan et al., 2012*). Because recent studies have suggested that non-coding RNAs associate with centromere and kinetochore proteins, including CENP-A, CENP-C and Aurora B (*Blower, 2016*; *Carone et al., 2009*; *Du et al., 2010*; *Ferri et al., 2009*; *Molina et al., 2017, 2016*; *Rošić et al., 2014*; *Wong et al., 2007*), it is possible that transcription of non-coding RNAs plays a role in centromere assembly and function. However, whether this occurs during mitosis or during G1, when centromere assembly takes place in mammalian cells (*Jansen et al., 2007*) remains unclear. Centromeric transcription is involved in nucleosome disassembly during interphase (*Boeger et al., 2003*) to facilitate the replacement of histone H3 by CENP-A (*McKittrick et al., 2004*; *Tagami et al., 2004*). Intriguingly, the observation that the elongating form of RNA Polymerase II is present at centromeres during mitosis (*Chan et al., 2012*) leave open the possibility that some level of local transcription of non-coding RNAs might still take place. However, it should be noted that a recent study in *S. pombe* indicated that RNA Polymerase II stalls at centromeric DNA and the level of stalling is directly proportional to the level of cnp1p (centromere-specific histone H3) nucleosome assembly (*Catania et al., 2015*).

While we were unable to experimentally exclude that non-coding RNAs are being transcribed from the centromere during a prolonged mitosis, we failed to detect any significant change in the total levels of α-satellite RNA after transcription inhibition, in agreement with previous reports (*Liu et al., 2015*). Indeed, experimental increase of α-satellite RNA levels were shown to reduce the binding of Sgo1 to nucleosomes (*Liu et al., 2015*) and led to problems in chromosome segregation (*Chan et al., 2017*; *Zhu et al., 2011*), suggesting that the levels of α-satellite RNA must be tightly regulated to ensure a faithful mitosis.

Overall, our results demonstrate that, regardless of our capacity to detect residual transcription that might still take place on specific chromosomal loci during mitosis, this is not required for human cells to sustain, progress, exit or die in mitosis.

# Materials and methods

**Key resources table**

| Reagent type | Designation | Source or reference | Identifiers | Additional information |
| --- | --- | --- | --- | --- |
| Cell line (human) | HeLa parental | Provided by Y. Mimori-Kiyosue | | |
| Cell line (human) | GFP-Aurora B HeLa | Provided by M. Lampson | | |
| Cell line (human) | Cyclin B1-Venus HeLa | Provided by J. Pines | | |
| Antibody | anti-Aurora B pT232 (Rabbit Polyclonal) | Rockland Immuno chemicals | Cat#600-401-677 | IF: (1:500) |
| Antibody | anti-Aim1 (Mouse Monoclonal) | BD Biosciences | Cat#611083 | IF: (1:500) WB: (1:1,000) |
| Antibody | anti-Mad1 (Mouse Monoclonal) | Merck Millipore | Cat#MABE867 | IF: (1:500) |
| Antibody | anti-phospho-Histone H3 Ser10 (Mouse Monoclonal) | Abcam | Cat#ab14955 | IF: (1:100,000) |

*Continued on next page*

*Continued*

| Reagent type | Designation | Source or reference | Identifiers | Additional information |
|---|---|---|---|---|
| Antibody | anti-Cleaved Caspase-3 (Rabbit Polyclonal) | Cell Signaling Technology | Cat#9661 | IF: (1:1,000) |
| Antibody | anti-Centromere antibody (Human) | Fitzgerald | Cat#90C-CS1058 | IF: (1:500) |
| Antibody | anti-phospho-Mps1 Thr676 (Rabbit Polyclonal) | a gift from G. Kops | | IF: (1:2,000) |
| Antibody | anti-phospho-Histone H3 Thr3 (Mouse Monoclonal) | a gift from J. Higgins | | IF: (1:1,000) |
| Antibody | anti-HEC1 (Mouse Monoclonal) | a gift from R. Gassmann | | IF: (1:2,000) |
| Antibody | anti-KNL1 (Rabbit Polyclonal) | a gift from R. Gassmann | | IF: (1:500) |
| Antibody | anti-phosho DSN1 (Rabbit Polyclonal) | a gift from I. Cheeseman | | IF: (1:1,000) |
| Antibody | anti-phospho KNL1 (Rabbit Polyclonal) | a gift from I. Cheeseman | | IF: (1:1,000) |
| Antibody | anti-CENPA (Mouse Monoclonal) | a gift from L. Jansen | | IF: (1:200) |
| Antibody | anti-DSN1 (Mouse Monoclonal) | a gift from A. Musacchio | | IF: (1:200) |
| Antibody | Alexa 488- or 568-or 647 secondaries | Invitrogen | | IF: (1:1,000) |
| Antibody | anti-Survivin (Rabbit Polyclonal) | Novus Biologicals | Cat#NB500-201 | WB: (1:1,000) |
| Antibody | anti-Borealin (Rabbit Polyclonal) | a gift from R. Gassmann | | WB: (1:1,000) |
| Antibody | anti-α-tubulin clone B-512 (Mouse Monoclonal) | Sigma-Aldrich | Cat#T5168 | WB: (1:5,000) |
| Antibody | anti-RNA Pol II S2 (Rabbit Polyclonal) | Abcam | Cat#ab5095 | WB: (1:1,000) |
| Antibody | anti-mouse or anti-rabbit | Jackson Immuno Research | | WB: (1:5,000) |
| Chemical compound, drug | Nocodazole | Sigma-Aldrich | Cat#M1404 | 1 µM |
| Chemical compound, drug | MG132 | EMD Millipore | Cat#133407-82-6 | 5 µM |

*Continued on next page*

*Continued*

| Reagent type | Designation | Source or reference | Identifiers | Additional information |
|---|---|---|---|---|
| Chemical compound, drug | Mps1-IN-1 | Provided by N. Gray | | 10 µM |
| Chemical compound, drug | RO3306 | Roche | Cat#SML0569 | 10 µM |
| Chemical compound, drug | Actinomycin D | Sigma-Aldrich | Cat#A9415 | 8 µM |
| Chemical compound, drug | α-amanitin | Sigma-Aldrich | Cat#A2263 | 2 µM |
| Chemical compound, drug | Triptolide | Sigma-Aldrich | Cat#T3652 | 1 µM |
| Chemical compound, drug | Cycloheximide | Sigma-Aldrich | Cat#01810 | 35.5 µM |
| Chemical compound, drug | Ethidium Bromide | Sigma-Aldrich | Cat#E8751 | 25 µM |
| Chemical compound, drug | ZM447439 (Aurora B inhibitor) | Selleckchem.com | Cat#S1103 | 3.3 µM |

## Cell culture and reagents

GFP-Aurora B expressing HeLa cells and Cyclin B1-Venus expressing HeLa cells were provided by M. Lampson and J. Pines, respectively. All cell lines including parental HeLa cells were grown in DMEM or L15 medium supplemented with 10% FBS (Invitrogen) and penicillin/streptomycin (100 IU/ml and 100 µg/ml) in a 37°C incubator with 5% $CO_2$. Microtubule depolymerization was induced by nocodazole (Sigma-Aldrich) at 1 µM for 2–16 hr, according to the experiment. To inhibit the proteasome, induce a metaphase arrest, and prevent exit due to a compromised SAC, cells were treated with 5 µM MG132 (EMD Millipore). For Mps1 inhibition, cells were treated with 10 µM Mps1-IN-1 (provided by N. Gray). For Aurora B inhibition, cells were treated with 3.3 µM ZM447439 for 4 hr prior to immunofluorescence. G2-enriched extracts were derived from cells incubated for 16 hr with the Cdk1 inhibitor RO3306 (Roche) at 10 µM, whereas mitotic extracts were obtained by shake-off upon nocodazole treatment. For transcription inhibition, actinomycin D, α-amanitin and triptolide (all from Sigma-Aldrich) were added to final concentration of 8 µM, 2 µM and 1 µM, respectively. To inhibit translation, cells were treated with 35.5 µM cycloheximide (Sigma-Aldrich). Ethidium bromide (Sigma-Aldrich) was used at final concentration of 25 µM. The cell lines were routinely tested negative for mycoplasma contamination.

## Live cell imaging

Control and transcriptionally repressed HeLa cells were imaged with phase-contrast microscopy (Axiovert 200M; Carl Zeiss; 20 × objective lens; A-Plan Ph1; 0.3 NA) equipped with a CCD camera (CoolSNAP HQ2; Photometrics) at 37°C in DMEM supplemented with 10% FBS. To determine mitotic timing, images were captured every 15 min for 72 hr using the Micro-Manager 1.3 software (www.micro-manager.org). In the washout experiment, images were captured every 20 min for 24 hr. HeLa cells stably expressing GFP-Aurora B (LAP-Aurora B) were imaged with an inverted microscope (TE2000U; Nikon; 20 × objective lens; LWD; 0.4 NA) equipped with an electron-multiplying charge-coupled device (CCD) camera (iXonEM+; Andor Technology) at 37°C in DMEM phenol red free medium (Invitrogen) supplemented with 10% FBS and 25 mM HEPES. Eleven 1 µm separated z-planes covering the entire volume of the mitotic spindle were collected every hour for 6 hr using the NIS-Elements Viewer software (Nikon). Cyclin B1-Venus were imaged every 15 min for 7 hr using an IN Cell Analyzer 2000 microscope (GE Healthcare) at 37°C in phenol red DMEM free medium supplemented with 10% FBS. The duration of all drug treatments in live-cell experiments are schematically illustrated in each respective figure. All images were analyzed with open source image analysis software ImageJ and cell profiler.

## Immunofluorescence microscopy

HeLa cells were fixed with 4% paraformaldehyde for 10 min and subsequently extracted in 0.3% Triton X-100 in 1 x PBS for 10 min. After short washes in PBS with 0.1% Triton X-100 and blocking with 10% FBS in PBS with 0.1% Triton X-100, all primary antibodies were incubated at 4°C overnight. Then, the cells were washed with PBS containing 0.1% Triton X-100 and incubated with the respective secondary antibodies for 1 hr at room temperature. DNA was counterstained with DAPI (1 μg/ml; Sigma-Aldrich) before coverslips were mounted in 90% glycerol +10% Tris, pH 8.5,+0.5% N-propyl gallate on glass slides. Rabbit anti-Aurora B pT232 (1:500; Rockland Immunochemicals Inc.), mouse anti-Aim1 (1:500; BD Biosciences), mouse anti-Mad1 (1:500; Merck Millipore), mouse anti-phospho-Histone H3 Ser10 (1:100,000; Abcam), rabbit anti-Cleaved Caspase-3 (1:1,000; Cell Signaling Technology), human anti-centromere antibody (ACA; 1:500, Fitzgerald Industries International), rabbit anti-phospho-Mps1 Thr676 (1:2000, a gift from G. Kops), mouse anti-phospho-Histone H3 Thr3 (1:1000, a gift from J. Higgins), mouse anti-HEC1 (1:2000, a gift from R. Gassmann), rabbit anti-KNL1 (1:500, a gift from R. Gassmann), rabbit anti-phosho-DSN1 and anti-phospho-KNL1 (1:1000, a gift from I. Cheeseman), mouse anti-CENPA (1:200, a gift from L. Jansen) and mouse DSN1 (1:200, a gift from A. Musacchio) were used as primary antibodies, and Alexa Fluor 488, 568, and 647 (Invitrogen) were used as secondary antibodies (1:1,000). Images were acquired on an AxioImager Z1 (63×, Plan oil differential interference contrast objective lens, 1.4 NA; all from Carl Zeiss) equipped with a charge-coupled device (CCD) camera (ORCA-R2; Hamamatsu Photonics) using the Zen software (Carl Zeiss) and blind deconvolved using Autoquant X (Media Cybernetics). Forty-one 0.2 μm separated z-planes covering the entire mitotic cell were collected. Images were analyzed in ImageJ, processed in Photoshop CS4 (Adobe) and represented with a maximum intensity projection of a deconvolved z stack.

## Fluorescence quantification

For quantitative measurements, all images compared were acquired using identical acquisition settings. Image analysis was performed using ImageJ. Briefly, individual kinetochores were identified by ACA staining and marked by a region of interest (ROI). The average fluorescence intensity of signals (pixel gray levels) of pAurora B at the inner centromere and Mad1 at kinetochores were measured on the focused z plan, and the background fluorescence was measured outside the ROI and subtracted. Fluorescence intensity measurements were normalized to the ACA signals. The results are reported after normalized to relative fluorescence levels in control samples. Approximately 1,300 KT pairs from 140 cells were analyzed for pAurora B and for Mad1, approximately 500 KT pairs from 50 cells were analyzed. The mean fluorescence intensity of LAP-Aurora B signals (pixel gray levels) from the live cell imaging was measured on the sum projection and the background fluorescence was measured outside the ROI and subtracted. Fluorescence intensities were normalized to the level at time = 0 and represented as a function of time. Approximately eight cells were analyzed per condition.

## Western blot

HeLa cells were ressuspended and lysed in NP-40 lysis buffer (20 mM HEPES-KOH pH 7.9, 1 mM EDTA, 1 mM EGTA, 150 mM NaCl, 0.5% (v/v) NP-40, 20% (v/v) Glycerol, 2 mM DTT, 1 mM PMSF) supplemented with protease and phosphatase inhibitors. The protein concentration was measured using the Bradford protein assay and 50 μg of proteins were resuspended in Protein Loading Buffer. Proteins were separated into a 4–20% SDS-PAGE and transferred to a nitrocellulose Hybond-C membrane using an iBlot Gel Transfer Device (Thermo Scientific). The membranes were blocked with 5% milk in PBS with 0.1% Tween-20 (PBST) at room temperature during 1 hr, and all primary antibodies were incubated at 4°C overnight. After five washes in PBST, the membranes were incubated with the secondary antibodies during 1 hr at room temperature. The membranes were washed in the same conditions than previously and the detection was performed with Clarity Western ECL Substrate (Bio-Rad). The following antibodies were used for western blot: rabbit anti-Survivin (1:1000; Novus Biologicals), mouse anti-Aim1 (1:1000; BD Biosciences), rabbit anti-Borealin (1:1000; provided by R. Gassmann), mouse anti-α-tubulin (clone B-512, 1:5,000; Sigma-Aldrich) and rabbit RNA Pol II S2 (1:1000; Abcam) were used as primary antibodies, and anti–rabbit and anti–mouse

antibodies were used as secondary antibodies (1:5,000; Jackson ImmunoResearch Laboratories, Inc.).

## RNA extraction and transcription analysis by RT-qPCR

Total RNA was isolated from interphase and mitotic HeLa cells (shake-off) using trizol (Life technologies). Reverse transcriptase reaction was performed with 500 ng of total RNA with iscript (Bio-rad) using the random primers and oligo(dTs) supplied, following the manufacturer's instructions. All primer sequences are in Supplemental Material. For each analysis, gapdh was used for normalization. RT-qPCRs were performed in the iCycler iQ5 Real-Time PCR Detection System (Bio-Rad Laboratories). The data obtained were analyzed using the Bio-Rad iQ5 Optical System Software v2.1 (BioRad Laboratories). The amplified products were additionally analyzed on agarose gel electrophoresis.

## RNA library preparation for RNAome sequencing

Quantity and total RNA integrity was checked following isolation using an Agilent Technologies 2100 Bioanalyzer. Sample with RNA Integrity Number (RIN) value greater than eight were selected. The RNA samples were prepared as described before in (*Derks et al., 2015*) with the following modifications. Five micrograms of total RNA were depleted according to the Illumina Ribo-zero magnetic protocol (www.illumina.com). The Ribo-zero-treated RNA was purified using Agencourt RNAClean XP Beads. RNA was fragmented on a Covaris S220. One microliter of the sheared rRNA depleted RNA was loaded on an Agilent Technologies 21000 Bioanalyzer RNA Pico chip to determine successful ribosomal RNA depletion. The sheared RNA sample was concentrated in a Thermo scientific SPD100 speedvac to 5 μl and a sequencing library was prepared according to the Illumina TruSeq Small RNA protocol (www.illumina.com). In short, adapters are subsequently ligated to the 3'end and the 5'end of the RNA. The RNA is reverse transcribed, amplified by PCR and run on SDS-PAGE gel. RNA fragments between 140 and 500 bp are cut out of the gel and purified. One microliter was loaded on an Agilent Technologies 2100 Bioanalyzer using a DNA 1000 assay to determine the library concentration and quality.

## Bridge amplification and sequencing by synthesis

Cluster generation was performed according to the Illumina TruSeq SR Cluster kit v2 (cBot) Reagents Preparation Guide (www.illumina.com). Briefly, six RNAome libraries were pooled together to get a stock of 2 nM. Five microliter of the 2 nM stock was denatured with NaOH, diluted to 11 pM and hybridized onto the flowcell. The hybridized products were sequentially amplified, linearized and end-blocked according to the Illumina Single Read Multiplex Sequencing user guide. After hybridization of the sequencing primer, sequencing-by-synthesis was performed using the HiSeq 2000 with a 36-cycle protocol. The sequenced fragments were denatured with NaOH using the HiSeq 2000 and the index-primer was hybridized onto the fragments. The index was sequenced with a six-cycle protocol.

## Bioinformatics analysis

Raw FASTQ files were aligned to the human genome (NCBI/build37.1) and annotated using Gencode version 25. De novo transcriptome assembly was performed using Gencode annotation and the StringTie assembler (*Pertea et al., 2015*). Mapping of sequence reads was performed using STAR aligner and transcript detection and quantification was performed using Cufflinks. Gene/transcript levels were compared between control and actinomycin D-treated samples using the ballgown program (*Frazee et al., 2014*).

## Data deposition

Processed RNA-seq have been deposited and can be consulted at ArrayExpress under accession number E-MTAB-6661 (http://www.ebi.ac.uk/fg/annotare/).

## CPC-CTM expression and purification

Full length (f.l.) Survivin was cloned as a 3C-cleavable His-GFP tagged protein in a pRSET vector (Thermo Fisher Scientific), f.l. Borealin was cloned as a TEV-cleavable His-tagged protein in a pETM

vector (gift from C. Romier, IGBMC, Strasbourg), and INCENP$_{1-58}$ was cloned as an untagged protein in a pMCNcs vector. The complex of f.l. Survivin, f.l. Borealin and INCENP$_{1-58}$ was prepared by co-expressing the subunits in *E. Coli* strain BL21 pLysS with an overnight induction at 18°C. Cells were lysed in lysis buffer containing 25 mM Hepes pH 7.5, 500 mM NaCl, 25 mM Imidazole, 2 mM β-mercaptoethanol (Bme). The complex was purified by affinity chromatography using a HisTrap HP column (GE Healthcare). The protein-bound column was washed with lysis buffer, followed by 25 mM Hepes pH 7.5, 1 M NaCl, 50 mM KCl, 10 mM MgCl, 25 mM Imidazole, 2 mM ATP, 2 mM Bme and proteins were eluted using 25 mM Hepes pH 7.5, 500 mM NaCl, 500 mM Imidazole, 2 mM Bme. Tags were cleaved during an overnight incubation with 3C and TEV while dialysing against 25 mM Hepes pH 7.5, 150 mM NaCl, 4 mM dithiothreitol (DTT) at 4°C. The complex was further purified by a cation exchange chromatography (HiTrap SP, GE Healthcare) followed by gel filtration using a Superdex 200 increase 10/300 column (GE Healthcare) equilibrated with 25 mM Hepes pH 8, 200 mM NaCl, 4 mM DTT.

## Expression and purification of recombinant histones and refolding of histone octamers

Human H2A and H2B and *Xenopus laevis* H3 and H4 were purified as described before (*Luger et al., 1999*, Methods Mol Biol) with minor changes. LB media was used instead of 2X TY-AC media for expression of H2A, H2B and H3 in *E. coli* BL21 (DE3) pLysS cells and H4 in *E. coli* BL21 cells. The preparation of inclusion bodies was performed using a Dounce glass/glass homogenizer. After solubilisation of the inclusion bodies and the three-step dialysis against Urea dialysis buffer (7 M Urea, 100 mM NaCl, 10 mM Tris pH 8, 1 mM EDTA, 5 mM Bme), the sample was applied to a HiTrap Q anion exchange column and then to a HiTrap SP cation exchange column (GE Healthcare). The histones were eluted from the HiTrap SP column using a linear gradient from 100 mM to 1 M NaCl in 7 M Urea, 10 mM Tris pH 8, 1 mM EDTA and 1 mM DTT. After the ion exchange step, purified recombinant histones were dialyzed against water with 5 mM Bme, lyophilized and stored at −80°C. Refolding of histone octamers was performed as previously described (*Luger et al., 1999*, Methods Mol Biol). Briefly, lyophilized core histones were resuspended in unfolding buffer (7M Guanidine HCl, 20 mM Tris pH 7.5, 10 mM DTT) and mixed to equimolar ratios. The histone mix was then dialyzed against three changes of 500 ml refolding buffer (10 mM Tris pH 8, 2M NaCl, 1 mM EDTA, 5 mM Bme) and the octamers were purified by size exclusion chromatography using a Superdex 200 increase 10/300 column (GE Healthcare) equilibrated with refolding buffer. Pooled fractions with equimolar ratios of histones were stored at −80°C.

## Nucleosome core particle reconstitution and drug treatment

The 147 bp 601 Widom positioning sequence was amplified from a pBS-601Widom vector with 5' IR700 labelled primers. Mononucleosome reconstitution was carried out using the salt gradient dialysis method (*Luger et al., 1999*, Methods Mol Biol). Histone octamers were added to DNA to a 0.9 molar ratio of octamer to DNA after adjusting the salt concentration to 2 M NaCl. The mixture was then dialysed against TE buffer (10 mM Tris pH 8, 1 mM EDTA, 50 mM NaCl) by gradually decreasing the ionic strength from 2 M to 50 mM NaCl over a period of 20 hr using a peristaltic pump. Reconstituted NCPs were treated with DMSO, 80 μm Actinomycin D, 10 μm Triptolide or 25 μm Ethidium Bromide for 1 hr at 4°C before incubating them with the CPC-CTM recombinant protein.

## Electrophoretic mobility shift assay

Electrophoretic mobility assay (EMSA) was used to detect specific interaction between CPC and the nucleosome core particles (NCPs). 25 and 50 nM recombinant CPC-CTM was added to 5 nM IR700-labelled NCPs in reaction buffer (25 mM Hepes pH 7.5, 100 mM NaCl, 1 mM DTT, 10% Glycerol). Reactions were incubated 30 min at 4°C and resolved in a 0.5% agarose gel in 1X Tris-CAPS buffer (60 mM Tris, 40 mM CAPS pH 9.3). The fluorescent bound and unbound NCPs were detected with Odyssey CLx Infrared Imaging System (LI-COR Biosciences). The ratio of CPC-bound NCPs to free NCPs was quantified using the Image Studio software (LI-COR Biosciences). Signal at the 0 nM CPC lane was subtracted from all values to account for any effect on NCPs destabilization upon drug treatment (EtBr-treated nucleosomes showed a faint smear above the unbound-NCP band indicating a slight NCP destabilization upon EtBr treatment).

## Statistical analysis

For statistical analysis, we first performed a proportion test or unpaired Student's t-test to confirm using p values that we have sampled a sufficient number of cells or kinetochores for concluding on the reported differences. Second, we measured standard error of the mean (SEM) across experimental repeats or across cells and confirmed that the reported differences were not only based on differences in mean values, but also the spread of the mean values between experiments, cells or kinetochores. SD values were obtained across experiments or cells as indicated in the figure legend. Statistical analysis was performed using Graphpad Prism version 5. p-Values were calculated with a Student's t test. For data that did not follow a normal distribution, statistical analysis was performed using a Mann-Whitney Rank Sum test.

## Acknowledgements

We thank Jonathan Pines (ICR, London, UK), Ben Black and Michael Lampson (University of Pennsylvania, Pennsylvania, USA) for providing the cyclin B1-venus and LAP-Aurora B HeLa cells, respectively, Reto Gassmann (i3S, Porto, Portugal), Iain Cheeseman (MIT, Cambridge, USA), Andrea Musacchio (MIP, Dortmund, Germany), Jonathan Higgins (Institute for Cell and Molecular Biosciences, Newcastle, UK), Lars Jansen (IGC, Oeiras, Portugal) and Geert Kops (Hubrecht Institute, Utrecht, The Netherlands) for the generous gift of antibodies, Nathanael Gray (Harvard University, USA) for the gift of Mps1-IN-1, Paula Sampaio (i3S, Porto, Portugal) and André Maia (i3S, Porto, Portugal) for assistance with microscopy, Patricia Oliveira and Pedro Ferreira from Bioinf2Bio for bioinformatics and data analysis, and Bernard Orr for the critical reading of the manuscript and helpful suggestions. We also thank José Silva for inspiring discussions and CID Lab members for feedback during the course of this project. The authors declare no conflicting financial interests. MC holds a doctoral fellowship (SFRH/BD/117063/2016) and CF has an Investigator starting grant (IF/00765/2014) from Fundação para a Ciência e a Tecnologia (FCT) of Portugal. This project has been funded by Norte-01–0145-FEDER-000029 and Norte-07–0124-FEDER-000003 from Norte 2020 and EXPL/IF/00765/2014/CP1241/CT0003 from FCT. Work in the laboratory of HM is funded by the European Research Council (ERC) under the European Union's Horizon 2020 research and innovation programme (grant agreement No 681443) and FLAD Life Science 2020.

## Additional information

### Funding

| Funder | Grant reference number | Author |
|---|---|---|
| European Regional Development Fund | COMPETE 2020: Norte-01-0145-FEDER-000029 | Cristina Ferrás |
| European Regional Development Fund | COMPETE 2020: Norte-07-0124-FEDER-000003 | Cristina Ferrás |
| Fundação para a Ciência e a Tecnologia | EXPL/IF/00765/2014/CP1241/CT0003 | Cristina Ferrás |
| Fundação para a Ciência e a Tecnologia | FCT Investigator grant IF/00765/2014 | Cristina Ferrás |
| Fundação para a Ciência e a Tecnologia | FCT PhD grant SFRH/BD/117063/2016 | Marco Novais-Cruz |
| European Research Council | Horizon 2020 research and innovation programme: 681443 | Helder Maiato |
| Fundação Luso-Americana para o Desenvolvimento | FLAD Life Science 2020 | Helder Maiato |

The funders had no role in study design, data collection and interpretation, or the decision to submit the work for publication.

## Author contributions
Marco Novais-Cruz, Investigation, Methodology, Writing—original draft; Maria Alba Abad, Investigation; Wilfred FJ van IJcken, Methodology; Niels Galjart, A Arockia Jeyaprakash, Investigation, Methodology; Helder Maiato, Conceptualization, Resources, Supervision, Funding acquisition, Investigation, Writing—review and editing; Cristina Ferrás, Conceptualization, Formal analysis, Supervision, Funding acquisition, Investigation, Methodology, Writing—original draft, Writing—review and editing

## Author ORCIDs
Wilfred FJ van IJcken (iD) http://orcid.org/0000-0002-0421-8301
Helder Maiato (iD) https://orcid.org/0000-0002-6200-9997
Cristina Ferrás (iD) https://orcid.org/0000-0003-1134-7387

## Decision letter and Author response
Decision letter https://doi.org/10.7554/eLife.36898.024
Author response https://doi.org/10.7554/eLife.36898.025

# Additional files

## Supplementary files
• Transparent reporting form
DOI: https://doi.org/10.7554/eLife.36898.020

## Data availability
Processed RNA-seq have been deposited and can be consulted at https://www.ebi.ac.uk/arrayexpress/experiments/E-MTAB-6661

The following dataset was generated:

| Author(s) | Year | Dataset title | Dataset URL | Database, license, and accessibility information |
|---|---|---|---|---|
| Ferrás C | 2018 | RNA seq Mitotic cells | https://www.ebi.ac.uk/arrayexpress/experiments/E-MTAB-6661 | Publicly available at ArrayExpress (accession no. E-MTAB-6661) |

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
