## [Decision Letter]

Thank you for submitting your article "Mitotic progression, arrest, exit or death is determined by centromere integrity, independently of de novo transcription" for consideration by *eLife*. Your article has been reviewed by three peer reviewers, including Jon Pines as the Reviewing Editor and Reviewer #1, and the evaluation has been overseen by Anna Akhmanova as the Senior Editor. The following individuals involved in review of your submission have agreed to reveal their identity:; Patrick Meraldi (Reviewer #3).

The reviewers have discussed the reviews with one another and the Reviewing Editor has drafted this decision to help you prepare a revised submission.

Summary:

In this study the authors have investigated whether transcription is required for exit from or arrest or death in mitosis. This is a topical and contentious issue and thus appropriate for *eLife*. The authors have carried out a thorough study and conclude that the compounds used to inhibit transcription perturb mitosis through their action to intercalate into centromeric DNA rather than through inhibiting transcription, and that compounds inhibiting RNA Pol II directly do not have the same effect. They further show that intercalation in centromeres perturbs mitosis by displacing Aurora B from centromeres.

Essential revisions:

1) Although the comparison of ActD and Triptolide is extremely interesting the analysis in Figure 4A suggests that transcription in interphase cells is not fully blocked. To draw any conclusions from negative data upon Triptolide inhibition it is extremely important to have 100% inhibition of transcription. It can be that full inhibition of transcription would actually give an effect even from a compound that does not intercalate DNA.

2) The reason behind mislocalization of CPC in ActD inhibited cells should be further investigated. Is H3T3p affected – a readout of Haspin activity – or is it the underlying nucleosome structure? Looking at marks for CPC localization in ActD would be worth doing. Alternatively, do intercalating agents change the kinetochore structure, in particular the SAC recruitment platform (KNL1 and Ndc80 staining).

3) While the authors show that intercalating agents lead to lower levels of phospho (active) Aurora-B and Mad1 at kinetochores, they do not prove that the faster mitotic slippage is directly caused by lower Aurora B levels at kinetochores. This is important, since it is not even clear in the field whether complete loss of Aurora B abolishes the SAC (it certainly leads to a delay in Mps1 recruitment).

To do this the authors could measure Aurora-B activity at centromeres/kinetochores in nocodazole treated cells +/- Actinomycin D with phospho-CENP-A or phospho-Dsn1 antibodies; titrate down Aurora-B activity to the same levels with the Aurora-B inhibitor ZM1, and test whether this is sufficient to weaken Mad1 recruitment and enhance mitotic slippage. Probing for non-phosphorylated Dsn1 at the same time would give a read out of kinetochore integrity. Alternatively, the authors could force Aurora-B at centromeres with an CENP-B-INCENP construct and ask whether this rescues the effect of intercalating agents. If the authors can rescue the cell fate effect by artificial recruitment of CPC to centromeres this would strongly support their model.

4) It would also be reassuring to repeat a key result with a non-transformed cell line.

---

## [Author Response]

Summary:In this study the authors have investigated whether transcription is required for exit from or arrest or death in mitosis. This is a topical and contentious issue and thus appropriate for eLife. The authors have carried out a thorough study and conclude that the compounds used to inhibit transcription perturb mitosis through their action to intercalate into centromeric DNA rather than through inhibiting transcription, and that compounds inhibiting RNA Pol II directly do not have the same effect. They further show that intercalation in centromeres perturbs mitosis by displacing Aurora B from centromeres.

We thank the reviewers for recognizing the appropriateness of our study for *eLife* and our effort to clarify a long-lasting controversy regarding the role of transcription during mitosis. We have now built on their constructive feedback to clarify with additional experiments all the issues raised in our original submission.

Essential revisions:1) Although the comparison of ActD and Triptolide is extremely interesting the analysis in Figure 4A suggests that transcription in interphase cells is not fully blocked. To draw any conclusions from negative data upon Triptolide inhibition it is extremely important to have 100% inhibition of transcription. It can be that full inhibition of transcription would actually give an effect even from a compound that does not intercalate DNA.

This is an excellent point and we thank the reviewers for bringing it up to our attention for further clarification. We would like to point out that the residual levels of Mcl1 in Figure 4A should not be taken tout court to infer about the percentage of inhibition of transcription in our experiments. The quantification presented in Figure 4A was performed after 4h in the presence of the drugs in an asynchronous cell population. This time was chosen because it corresponds approximately to the minimal time from which interphase cells start dying in the presence of the drugs. Thus, the residual levels of Mcl1 in Figure 4A likely reflect the asynchronous state of the cell population and mRNAs that had just been transcribed right before the addition of the drugs. Importantly, the residual levels of Mcl1 found after Triptolide treatment were equivalent to those found after Actinomycin D, making it unlikely that a different extent of transcription inhibition accounts for the observed differences in cell response. Finally, interphase cells start to die within 4-5 hours in the presence of either Triptolide or Actinomycin D, suggesting an efficient (and equivalent) inhibition of transcription in both scenarios. These results are now highlighted in the text. We have also now represented the quantification in terms of mean and standard deviation.

2) The reason behind mislocalization of CPC in ActD inhibited cells should be further investigated. Is H3T3p affected – a readout of Haspin activity – or is it the underlying nucleosome structure? Looking at marks for CPC localization in ActD would be worth doing. Alternatively, do intercalating agents change the kinetochore structure, in particular the SAC recruitment platform (KNL1 and Ndc80 staining).

We followed the reviewers’ suggestions and investigated the phosphorylation of histone H3 at threonine 3 (*H3T3p*) as a readout of Haspin activity, as well as several structural kinetochore components (and their Aurora B-mediated phosphorylation state). We found no statistically significant differences in phosphorylation on H3T3p, CENP-A, Hec1 (Ndc80 complex), Dsn1 (Mis12 complex) and active Mps1 (Mps1 pT676; please see next point) in cells treated with nocodazole, with or without Actinomycin D. However, we found a reduction of pDsn1 and pKnl1 in cells treated with nocodazole+Actinomycin D, in agreement with perturbation of Aurora B activity on centromeres. Importantly, while total Dsn1 was not affected by treatment with Actinomycin D, total Knl1 was. We concluded that, with the exception of Knl1 (please see next point), Actinomycin D treatment for 4 hours does not compromise other key structural kinetochore components. Moreover, the observed effect of Actinomycin D treatment over Aurora B localization at centromeres appears to be independent of Haspin and Mps1 activity (note also that, contrary to Actinomycin D treatment, Mps1 inhibition immediately abrogates the SAC).

3) While the authors show that intercalating agents lead to lower levels of phospho (active) Aurora-B and Mad1 at kinetochores, they do not prove that the faster mitotic slippage is directly caused by lower Aurora B levels at kinetochores. This is important, since it is not even clear in the field whether complete loss of Aurora B abolishes the SAC (it certainly leads to a delay in Mps1 recruitment).To do this the authors could measure Aurora-B activity at centromeres/kinetochores in nocodazole treated cells +/- Actinomycin D with phospho-CENP-A or phospho-Dsn1 antibodies; titrate down Aurora-B activity to the same levels with the Aurora-B inhibitor ZM1, and test whether this is sufficient to weaken Mad1 recruitment and enhance mitotic slippage. Probing for non-phosphorylated Dsn1 at the same time would give a read out of kinetochore integrity. Alternatively, the authors could force Aurora-B at centromeres with an CENP-B-INCENP construct and ask whether this rescues the effect of intercalating agents. If the authors can rescue the cell fate effect by artificial recruitment of CPC to centromeres this would strongly support their model.

This question has been partially answered in the previous point (i.e. Mps1 recruitment, as well as total and pDsn1). In addition, because we found a reduction in total Knl1 upon Actinomycin D treatment and Knl1 is known to recruit several SAC proteins (including Mad1) to kinetochores, we tested whether Knl1 and Mad1 recruitment to kinetochore depends on Aurora B activity at the centromeres. To do so we investigated total Knl1 and Mad1 levels at kinetochores upon Aurora B inhibition with ZM447439. We found that Aurora B inhibition caused a proportional reduction in Knl1 and Mad1, comparable with Actinomycin D treatment. Together with our in vitro reconstitution data, these data strongly suggest that perturbation of Aurora B activity at centromeres is the primary effect resulting from DNA intercalation by Actinomycin D and this likely accounts for the weakened SAC response due to a downstream effect over Knl1 and Mad1 recruitment. These results are now reported in Figure 3—figure supplement 2 and are further supported by the suggested rescue experiment with a CENP-B-INCENP construct that was able to overcome the SAC deficiency caused by Actinomycin D by constitutively targeting Aurora B to centromeres. This experiment had already been performed by Bastians and colleagues (Becker et al., 2010) and is cited in our manuscript.

4) It would also be reassuring to repeat a key result with a non-transformed cell line.

We have now treated RPE1 cells with Actinomycin D and confirmed the mislocalization of active Aurora B from centromeres under these conditions by immunofluorescence. This result is now shown in Figure 2—figure supplement 3A,B. We have also tried hard to measure the duration of mitosis and fate after mitotic arrest in RPE1 cells upon nocodazole treatment, with and without Actinomycin D, but tracking of these cells under these conditions was extremely difficult due to their high motility, detachment and clustering. For these reasons, we prefer not to draw any definitive conclusion in this system.